

# Measurements of C10-C15 biogenic volatile organic compounds (BVOCs) with sorbent tubes

Heidi Hellén[1], Toni Tykkä[1], Simon Schallhart[1], Evdokia Stratigou[2], Thérèse Salameh[2], Maitane Iturrate-Garcia[3]

[1]Atmospheric Composition Research, Finnish Meteorological Institute, 00560 Helsinki, Finland

[2]IMT Nord Europe, Institut Mines-Télécom, Univ. Lille, Centre for Energy and Environment, F-59000, Lille -France

[3]Department of Chemical and Biological Metrology, Federal Institute of Metrology (METAS), Bern-Wabern 3003, Switzerland

*Correspondence to*: Heidi Hellén (heidi.hellen@fmi.fi)

**Abstract.** Biogenic Volatile Organic Compound (BVOCs; e.g. terpenes) are highly reactive compounds with very low amount fractions in the air. Due to this, their measurements are challenging and they may suffer losses during sampling, storage and analyses. In this study, the performance of an off-line technique for the measurement of BVOCs based on sorbent tubes sampling was evaluated. Even though online measurements of BVOCs are becoming more and more common, the use of

sorbent tubes is expected to continue since they have greater spatial coverage compared to online GC measurements and no infrastructure is needed for sampling. Tested compounds included 8 monoterpenes, 5 sesquiterpenes and 5 oxy BVOCs, which are generally either directly emitted (1,8-cineol, linalool, bornylacetate) or oxidation products (nopinone and 4-methylacetylcyclohexene). Two sorbent materials (Tenax TA and Carbopack B) and four tube materials (stainless steel (SS), SilcoNert 1000, glass and glass coated SS) were used. The laboratory evaluations determined storage stability, breakthrough

volumes, suitable tube materials, recoveries from ozone scrubbers and particulate filters and the sampling efficiency. In addition, an intercomparison between two laboratories was conducted.

Of the sorbent materials Tenax TA showed acceptable results for these BVOCs, while with Carbopack B losses and increases of some compounds were detected. Studied compounds were found to be stable in Tenax TA tubes for at least one month at ±20 °C. Breakthrough tests indicated that α- and β-pinene have clearly lower breakthrough volumes in Tenax TA (4–7 and 8–

26 L, respectively) than other terpenes (>160 L). SS, SilcoNert 1000 and glass were all shown to be suitable tube materials. Results from Tenax TA sorbent tube sampling agreed with online sampling for most compounds. Heated SS tubes, sodium thiosulfate filters and KI/Cu traps were found to be suitable ozone scrubbers for studied BVOCs. Tested particle filters had a greater impact on limonene than on α- and β-pinene. The laboratory intercomparison of α- and β-pinene measurements showed that in general, measured values by the two laboratories were in good agreement with Tenax TA.





## 1. Introduction

Biogenic emissions of volatile organic compounds (VOCs) are globally ~10 times higher than anthropogenic ones (Guenther et al., 2012). The major group of biogenic volatile organic compounds (BVOCs) in the atmosphere is terpenoids. They are mainly emitted from the vegetation (Sinderalova et al., 2014), but in urban areas also volatile chemical products e.g. hygiene and cleaning products can be their sources (Hellén et al., 2012a, Steinemann, 2015; Coggon et al., 2021). A recent study by Borbon et al. (2023) indicates that even tailpipe exhausts may be a significant source of terpenoids especially in cities of the developing world. Terpenes are classified according to the number of isoprene ($C_5H_8$) units into monoterpenes ($C_{10}H_{16}$), sesquiterpenes ($C_{15}H_{24}$) and diterpenes ($C_{20}H_{32}$). While some forests emit mainly isoprene, boreal forests, for example, are strong mono- and sesquiterpene emitters (e.g. Schallhart et al., 2018; Hakola et al., 2006, 2017; Hellén et al., 2018, 2021).

Terpenes are highly reactive with respect to atmospheric oxidants such as ozone ($O_3$), the hydroxyl radical (OH) and the nitrate radical ($NO_3$) and their atmospheric lifetimes vary from a few minutes to several hours (Navarro et al., 2014). Terpenes affect greatly the oxidative capacity of the atmosphere and the formation and destruction of ozone. They also participate in the formation and growth of new particles and clouds, impacting climate. Even small changes in BVOC emissions may substantially modify the radiative properties of clouds (Petäjä et al., 2022). Furthermore, most of the terpenes emitted have a therapeutic potential for inflammatory diseases as result of their anti-inflammatory effects as well as their function against oxidative stress (Cho et al., 2017; Kim et al., 2020).

Atmospheric mono- and sesquiterpenes have been studied both using direct mass spectrometric (MS) and chromatographic methods. Proton transfer reaction mass spectrometers (PTR-MSs/PTR-time-of-flight (TOFs)) have been used for measurements of monoterpene ambient air amount fractions (e.g. Peräkylä et al., 2014) and emissions (e.g. Schallhart et al., 2018). More recently Vocus-PTR-TOFs have been used also for measurements of sesquiterpenes (Li et al., 2020, 2021). With these direct mass spectrometric techniques high time resolution (even seconds) can be achieved, but they lack the species-specific information and all monoterpenes or sesquiterpenes are detected as a sum. Due to variable reactivities of terpenes with respect to atmospheric oxidants, detailed information is needed for estimating their atmospheric impacts.

In addition to these direct MS methods, in situ chromatographic methods, which enable species specific detection of terpenes, have been used for BVOC measurements of ambient air (e.g. Bouvier-Brown et al., 2009; Hakola et al., 2012; Hellén et al., 2012, 2018, 2020; Yee et al., 2018; Mermet et al., 2019, 2021) and emissions (e.g. Hakola et al., 2017; Hellén et al., 2020, 2021). These in situ methods need good infrastructures with stable room temperature and electricity. Therefore, also sorbent tubes, which can be easily transported to several locations with low or no infrastructure, are used. With sorbent tubes, one analytical instrument in the laboratory can be used for analyzing samples from several different locations. Tubes are also re-usable. Therefore, sorbent tube sampling is a cost-effective alternative for BVOC sampling, especially on locations where in situ measurements are not possible. However, time resolution of sorbent tubes is often poor and sampling is more laborious compared to in situ "online" measurements.



65    Sorbent tubes are commonly used for offline sampling of VOCs in the air, both for ambient air and emission studies. While

there are standard and reference methods available for sorbent tube sampling of aromatic hydrocarbons (e.g. ISO/DIS 16017-

2 and EN 14662:2005) emitted by the different anthropogenic sources (e.g. traffic and biomass burning), much less data is

available on their suitability for different BVOCs. Most commonly sorbent, used in the ambient air studies of terpenes, has

been Tenax TA (e.g. Hakola et al., 2003; Jardine et al., 2015; Zannoni et al., 2016). In addition, carbon-based sorbents e.g.

Carbotrap (Gallego et al., 2010), Carbopack B (Oh et al., 2010), Carbograph 1 and 5 (Yanez-Serrano et al., 2018; Song et al.,

2012) and Chromosorb 106 (Sunesson et al., 1999) have been used in the ambient air, lab or emission studies. Often Tenax

TA is used together with other stronger sorbents to widen the selection of the compounds, which can be measured quantitatively

(e.g. Helin et al., 2020; Jardine et al., 2015; Hakola et al., 2003; Zannoni et al., 2016).

There are several different types of sorbents and tube materials available and their suitability depends greatly on the studied

compounds. Therefore, suitability of used materials should be evaluated for all compounds of interest. Tests with sorbent tubes

for analysis of some monoterpenes have been conducted by e.g. Komenda et al. (2001), Arnts (2010), Gallego et al. (2010),

Ullah and Kim (2014), Ahn et al. (2016) and Sheu et al. (2018), but very little information is available on their suitability for

sesquiterpene or $C_{10}$–$C_{15}$ oxygenated BVOC (oxy BVOC) measurements (Helmig et al., 2004; Helin et al., 2020).

Since most of the studied BVOCs are highly reactive with ozone ($O_3$) and can be oxidized also during sampling on the sorbent

tubes, ozone scrubbers are used to remove ozone in front of them. Several different $O_3$ removal techniques have been used in

VOC measurements e.g. $MnO_2$ nets, sodium thiosulfate impregnated filters, heated stainless steel (SS) tubes, copper tubes

coated with potassium iodide and NO titration (Frick et al., 2001; Pollman et al., 2005; Hellén et al., 2012b) . Some of the $O_3$

scrubbers are known to have very short lifetime (Frick et al., 2001; Bouvier-Brown et al., 2009) while some may suffer losses

of most reactive compounds (Calogirou et al., 1996; Pollman et al., 2005). In addition to scrubber material or type, optimizing

size and flow is also critical for achieving sufficient $O_3$ removal without losing studied BVOCs (Hellén et al., 2012b).

Filters are used to remove particulate matter in sorbent tube measurements and analyses. Particle filters are recommended to

avoid any contamination of all the fluidic parts of a measurement system, especially the more sensitive ones such as valves,

measurement cells, pre-concentration-traps and reactors. Guidelines mostly recommend polytetrafluoroethylene (PTFE) or

perfluoroalkoxy (PFA) membranes as well as stainless steel screens as particle filters (e.g. Steinbrecher and Weiß, 2012;

Reimann et al., 2018). VOCs are not trapped by these filters, but they may suffer on some losses.

Here the suitability of sorbent tubes for $C_{10}$–$C_{15}$ BVOC measurements was evaluated in the framework of the EMPIR

(European Metrology Programme for Innovation and Research) project *Metrology for Climate Relevant Volatile Organic

Compounds* (MetClimVOC, https://www.metclimvoc.eu/). Tested compounds included 8 monoterpenes, 5 sesquiterpenes and

5 oxy BVOCs, which are generally either directly emitted (1,8-cineol, linalool, bornylacetate) or oxidation products (nopinone

95    and 4-methylacetylcyclohexene). The laboratory evaluations determined storage stability, breakthrough volumes, suitable tube

materials, recoveries from $O_3$ scrubbers and particulate filters and the sampling efficiency. In addition, an intercomparison

between two laboratories was conducted. Based on the earlier BVOC studies, two sorbents (Tenax TA and Carbopack B, 60-

mesh) were selected for the laboratory evaluations.



## 2. Experimental

### 2.1 Chemicals and materials

Methanol (≥99.9 %) for producing terpene standards was purchased from VWR Chemicals (Gliwice, Poland) and pure solutions of the studied compounds (listed in Table 1) were purchased from Sigma-Aldrich (St.Loius, MO, USA). Tested compounds included 8 monoterpenes (α-pinene, camphene, myrcene, β-pinene, 3Δ-carene, p-cymene, limonene, terpinolene),

5 sesquiterpenes (longicyclene, iso-longifolene, β-farnesene, β-caryophyllene, α-humulene), 5 oxy BVOCs (1,8-cineol, linalool, 4-acetylmethylcyclohexane (4AMCH), nopinone and bornylacetate) as well as 3 aromatic hydrocarbons (toluene, o-xylene and 1,3,5-trimethylbenzene) for the comparison. Each compound was weighed in 500 mL of methanol, which was further diluted into six different amount-of-substance fractions (a.k.a. amount fractions). For producing VOC rich air, the methanol solution containing 10–15 µg mL$^{-1}$ of studied terpenes was injected to the zero air via a Teflon PTFE T-piece by

using an automatic syringe pump (at injection flow rate of 6–15 µL h$^{-1}$). Zero air used in the tests was generated using a zero air generator (HPZA-7000, Parker Balston, Lancaster, NY, USA). The air was humidified for desired level by bubbling a fraction of air through the ultrapure water (Milli-Q Gradient, Molsheim, France).

Tested tubes were commercial Tenax TA (60/80), self-packed Tenax TA (60/80) and commercial Carbopack B (60/80, see Table S1 for more details). Carbopack B has been used also in the cold traps of online online thermal desorption-gas

chromatography-mass spectrometers (TD-GC-MSs) (e.g. Mermet et al., 2019) and in multibed sorbent tubes together with Tenax TA (e.g. Ullah and Kim 2014; Iqbal et al., 2014; Helin et al., 2020). The main tube material was stainless steel (SS), which has been the most common material also in earlier studies. In addition, a set of self-packed SilcoNert 1000 tubes with Tenax TA were used. Even though SilcoNert 1000 is not commonly used material in sorbent tube sampling, it is known to be more inert material compared to SS and its use could improve the results.

The self-packed sorbent tubes were prepared by packing empty stainless-steel or SilcoNert 1000 tubes (PerkinElmer Inc., Waltham, MA, USA) with Tenax TA (60/80) purchased from Sigma-Aldrich (St. Louis, MO, USA). Silanized glass wool (Phase Separations Ltd., Deeside, UK), stainless-steel mesh (Markes International, Llantrisant, UK) and gauze-retaining spring (Markes International, Llantrisant, UK) were used to prevent sorbent phase mixing and exiting.

Before the sampling, the used sorbent tubes were conditioned by heating to 320 oC with 50 mL min-1 flow of helium for 15

minutes using a thermal desorption unit (TurboMatrix 350, Perkin-Elmer).




**Table 1**. List of the studied compounds with limits of quantification (LOQ) for Tenax TA tubes and uncertainty for Tenax TA ($U_T$) and Carbopack B ($U_C$) tubes. LOQ and U (k=2) were calculated for the sampling volume of 1.5 L and amount of ~30 ng of each studied compound. The equations for calculating uncertainties can be found in the Supplement S1.

| Compound | Class | CAS number | Molecular formula | LOQ (pg) | LOQ (pmol mol$^{-1}$) | $U_T$ (%) | $U_C$ (%) |
|---|---|---|---|---|---|---|---|
| α-Pinene | Monoterpene | 7785-70-8 | C10H16 | 30 | 3.6 | 7 | 10 |
| Myrcene | Monoterpene | 123-35-3 | C10H16 | 80 | 9.6 | 7 | 10 |
| β-Pinene | Monoterpene | 19902-08-0 | C10H16 | 30 | 3.6 | 7 | 25 |
| Terpinolene | Monoterpene | 586-62-9 | C10H16 | 70 | 8.4 | 10 | 38 |
| Camphene | Monoterpene | 79-92-5 | C10H16 | 30 | 3.6 | 8 | 18 |
| 3-Carene | Monoterpene | 498-15-7 | C10H16 | 20 | 2.4 | 7 | 10 |
| p-Cymene | Monoterpene | 99-87-6 | C10H14 | 20 | 2.4 | 7 | 8 |
| Limonene | Monoterpene | 5989-54-8 | C10H16 | 60 | 7.2 | 8 | 18 |
| Longicyclene | Sesquiterpene | 1137-12-8 | C15H24 | 40 | 3.2 | 7 | 30 |
| Isolongifolene | Sesquiterpene | 1135-66-6 | C15H24 | 40 | 3.2 | 7 | 42 |
| β-Caryophyllene | Sesquiterpene | 87-44-5 | C15H24 | 110 | 8.8 | 7 | 42 |
| β-Farnesene | Sesquiterpene | 18794-84-8 | C15H24 | 200 | 16 | 7 | 74 |
| α-Humulene | Sesquiterpene | 6753-98-6 | C15H24 | 30 | 2.4 | 7 | 60 |
| 1,8-Cineol | Oxy BVOC | 470-82-6 | C10H18O | 30 | 3.2 | 7 | 8 |
| Linalool | Oxy BVOC | 78-70-6 | C10H18O | 90 | 9.5 | 8 | 78 |
| 4AMCH* | Oxy BVOC | 6090-09-1 | C9H14O | 120 | 14 | 7 | 34 |
| Nopinone | Oxy BVOC | 38651-65-9 | C9H14O | 30 | 3.5 | 8 | 16 |
| Bornylacetate | Oxy BVOC | 5655-61-8 | C12H20O2 | 50 | 4.2 | 7 | 74 |
| Toluene | Aromatic HC** | 108-88-3 | C7H8 | 80 | 14 | 7 | 7 |
| o-Xylene | Aromatic HC** | 95-47-6 | C8H10 | 20 | 3.1 | 7 | 7 |
| 135TMB*** | Aromatic HC** | 108-67-8 | C9H12 | 20 | 2.7 | 7 | 8 |

*4AMCH=4-acetyl-1-methylcyclohexene, **HC=hydrocarbon, ***135TMB=1,3,5-trimethylbenzene

## 2.2 Analysis

The sorbent tubes from the BVOC tests were analyzed using a thermal desorption unit (TD, TurboMatrix 350, Perkin-Elmer) connected to a gas chromatograph (GC, Clarus 680, Perkin-Elmer) coupled to a mass spectrometer (MS, Clarus SQ 8 T, Perkin-Elmer). These tubes were desorbed in a helium flow of 50 mL min-1 at 300 °C. The sample was focused into a Tenax TA cold



trap kept at 20 °C. From there, the sample was transferred to the GC through a heated transfer line by rapid heating up to 300 °C in a helium flow. A DB-5 column (length 60 m, internal diameter (id.) 0.25 mm, film thickness 1 µm, from Agilent

Technologies) was used for separation. The GC column oven temperature was initially at 50 °C from which it was increased to 150 °C at a rate of 4 °C min-1 and then at a rate of 8 °C min-1 to 280 °C where it was kept for 8.75 min. Helium carrier gas flow was 1 mL min-1. The method has been described in Helin et al. (2020).TD-GC-MS was calibrated using methanol solutions for BVOCs. Methanol solutions (5 µL) of studied VOCs were injected into the sorbent tubes using a flow ~80 mL min-1 of nitrogen with quality 99.9999 % (50 L cylinder, Linde gas). Methanol was flushed away for 10 minutes. The final

tubes contained 0.25–78.9 ng of studied compounds depending on the calibration level. Six level calibration was used. Both Tenax TA tubes and multibed tubes with Tenax TA and Carbopack B were used for calibrating the instrument. For the comparison of calibration methods, a gaseous standard from National Physical Laboratories (NPL, UK) containing 2 nmol mol-1 α-pinene, 3Δ-carene, 1,8-cineol and limonene was used with dual stage valve (R200/2-6SS, Aga inc.). The gaseous standard was sampled both into the sorbents of the tubes and directly to the cold trap of TD. Sorbent tubes were sampled

directly from the outlet of the gas cylinder valve and for the direct measurements FEP tubing (length ~0.5m, id. 1/16 inch) was used.

The blanks were below detection limit for all studied compounds except toluene. For toluene, blank amounts of 0.47 ng and 0.22 ng were found in Tenax TA and Carbopack B tubes, respectively. The quantification limits presented in Table 1 were calculated as 10 times the signal-to-noise ratio and they varied between 20 and 200 pg/tube.

The uncertainty was estimated for the sampling volume of 1.5 L and amount of ~30 ng of each studied compound. The calculation of the total uncertainty is described in detail in Supplement S2. The uncertainty of the studied compounds in Tenax TA and Carbopack B tubes was 7–10 % and 7–78 %, respectively (Table 1).

Additional TD-GC-flame ionization detector (FID) was used for particulate filter studies.


**2.3 Storage stability tests**

The storage stability was tested by injecting known amounts (25-39 ng) of the studied compounds into the sorbent tubes as methanol solutions similarly to the preparation of calibration tubes. The tubes were sealed with brass Swagelok caps and PTFE ferrules. One third of the tubes (i.e. 4 commercial and 5 self-packed SS Tenax TA, 3 self-packed SilcoNert 1000 Tenax TA

and 5 commercial Carbopack B) were analyzed immediately, one third was kept at room temperature (~22 °C) for one month and one third was kept at -20 °C for one month before the analysis.

Relative difference of terpenes in the sorbent tubes kept at a) +22 °C and b) -20 °C for one month compared to the tubes analyzed immediately were calculated using Eq. (1):

$$Rel.diff_i = \frac{m_i - m_0}{m_0} \cdot 100\% \qquad (1)$$



where $m_i$ is the mass (ng) of studied compounds in the tubes after storage period of i and $m_0$ is the mass (ng) of studied compound in the tubes analyzed immediately after the preparation.

**2.4 Breakthrough volumes**

To determine safe sampling/breakthrough volumes of BVOCs, breakthrough tests were conducted. BVOCs were injected into the zero air flow to produce BVOC rich air with amount fractions of 0.2–10 nmol mol$^{-1}$. Most of the tests were conducted with methanol solutions of BVOCs injected to the zero air. However, two additional tests were conducted for α- and β-pinene using a portable generator of reference gas mixtures (ReGaS2; Pascale et al., 2017) – based on the permeation method (ISO

6145-10:2002) – to check if methanol injected with BVOCs to the zero air affects the breakthrough volumes.

Sorbent tubes were attached onto the inlet of the instrument and the amount fraction levels breaking through the tubes were followed online with the TD-GC-MS. Breakthrough (B) was calculated following the Eq. (2):

$$B = \frac{X_{out}}{X_{in}} \cdot 100\% \qquad (2)$$


where $X_{in}$ is the amount fraction of VOCs in the air flushed into the tube and $X_{out}$ is the amount fraction in the air coming out from the tube.

Breakthrough volumes were defined as a volume when 5 % of the injected amount fraction passes through the tube i.e., the volume when B = 5 % (EN 14662-1).


**2.5 Sampling efficiency**

To study the suitability of the used calibration method, a comparison with reference gas from NPL containing 2 nmol mol$^{-1}$ α-pinene, 3Δ-carene, 1,8-cineol and limonene was conducted by calibrating the instrument with methanol solution and analyzing the reference gas with the online mode of the instrument. Three different sampling times 10, 20 and 30 minutes were used with

a flow of 40 mL min$^{-1}$. Expected sample masses were 4.3–4.7 ng, 8.5–9.3 ng and 12.8–14 ng for 10, 20 and 30 minute sampling, respectively. Three replicate samples were taken with each sampling time.

The relative differences between the amount fractions of studied compounds in the sorbent tubes compared to the expected amount fractions were calculated using the Eq. (3):

$$Rel.\,diff_j = \frac{m_j - m_e}{m_e} \cdot 100\% \qquad (3)$$


where $m_j$ is the mass (ng) of studied compound in the tubes (j) and $m_e$ is the expected mass (ng).



Linearity of the response related to the sampling amount was used as one indication of the sampling efficiency. Sampling efficiency of the sorbent tubes was also studied by comparison with online TD-GC-MS results. Studied compounds were

injected as methanol solutions into the zero air flow and amount fraction levels were measured in parallel with different sorbent tubes and an online TD-GC-MS. Both commercial and self-packed, SS and SilcoNert 1000 Tenax TA tubes as well as commercial Carbopack B tubes were tested. Sampling efficiency of the sorbent tubes was estimated by sampling different amounts/volumes and by comparing the results to online TD-GC-MS results. The flow used for sorbent tube sampling was ~200 mL min$^{-1}$. Tests were conducted at two different relative humidities (RH ~30 % and ~70 %). Sampled volumes for tubes

were 3, 6 and 12 L and for the online sampling 0.6, 1.2 and 2.4 L. Sampled amounts were 2–80 ng/tube. Since no blank was detected in blank sorbent tube tests, zero was used as one point. Recovery of the sorbent tubes was compared to the recovery of the online sampling.

An additional test of linearity was conducted, using only commercial Tenax TA tubes, by taking five different sample volumes (0.4, 0.6, 0.8, 1.0 and 1.2 L) from ReGaS2 outlet. Amount fractions obtained by sampling different volumes were compared.

The sampled amount of α-pinene in the tubes varied from 14 ng to 44 ng. The commercial Tenax TA tubes were also compared with online TD-GC-MS while sampling from the ReGaS2 permeator. Both tubes and online samples were taken for 15 minutes with a flow of 40 mL min$^{-1}$ (sample volume 0.6 L).

Additional sampling efficiency tests were conducted for α-pinene, 3Δ-carene, 1,8-cineol and limonene with the 2 nmol mol$^{-1}$ reference standard gas (NPL, UK). Samples were taken from the gas into the commercial and self-packed Tenax TA and

commercial Carbopack B tubes with the sampling flow of ~100 mL min$^{-1}$ and sampling time of 10 min. Three replicates were taken of each type of tube. Samples were analyzed with the TD-GC-MS using the calibration with methanol solutions, which was also used in the other experiments and detected amounts were compared to expected amounts. At the same time samples were taken using online mode of the TD-GC-MS.

**2.6 Tests with different tube materials**

To study the impact of sorbent tube materials on recoveries, BVOC rich air was flushed through empty sorbent tubes made of SS, glass coated SS, glass and SilcoNert 1000 and the relative difference to the situation where empty tube was not used in front of sorbent tube was followed (Eq. 4, Fig. 1). To produce terpene rich air for the test, studied compounds were injected as methanol solutions into the zero air flow. Sorbent tube samples were taken from the terpene rich zero air flow with (sorbent

tube 2) and without (sorbent tube 1) empty tube attached in front (Fig. 1). Sampling time was 30 minutes and the flow 100 mL min$^{-1}$. Sampled masses of individual compounds were ~14-23 ng/tube. Relative difference between masses found in the sorbent tube 1 ($m_1$) and 2 ($m_2$) was calculated following the Eq. (4):

$$Rel.\,diff_x = \frac{m_2 - m_1}{m_1}\, x\, 100\% \qquad (4)$$






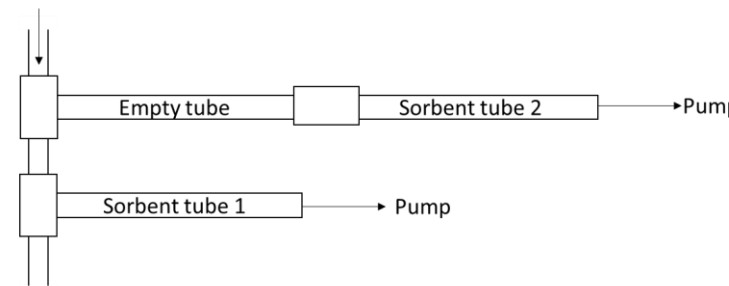

**Figure 1.** Set up for studying losses on the wall of the sorbent tubes.


## 2.7 O$_3$ scrubber tests

To determine the effect of different O$_3$ scrubbers on amount fractions of studied BVOCs methanol solutions of them were injected into zero air flow and measured upstream and downstream the scrubbers with and without ozone. Four O$_3$ scrubber

types were used; a heated stainless steel tube (heated SS) at 120 °C, a filter impregnated with thiosulfate (Na$_2$SO$_3$), manganese-oxide (MnO$_2$) net and copper tube coated with potassium iodide (KI/Cu). Table 2 gives details of the used scrubbers. The used ozone level was ~40 nmol mol$^{-1}$. Flows through the scrubbers were set at 1.0, 0.1, 0.08 and 0.1 L min$^{-1}$ for KI/Cu, MnO$_2$, Na$_2$SO$_4$ and heated SS-scrubbers, respectively. New and aged scrubbers were tested. This procedure was performed at relative humidity values 30–36 % and at room air temperature (~22 °C). A detailed schematic of the setup is depicted in supplement

Fig. S1.

Sorbent tube (Tenax TA 60-80/Carbopack B 60-80) sampling was conducted with a flow of ~80 mL min$^{-1}$ and sampling time of 15–60 minutes. Simultaneous samples were always taken upstream and downstream the scrubbers. Relative difference (Rel.diff$_o$) of masses found in the samples with (m$_w$) and without (m$_{wo}$) O$_3$ scrubbers was calculated from Eq. (5):


$$Rel.diff_o = \frac{m_w - m_{wo}}{m_{wo}} \ x \ 100\% \qquad (5)$$





**Table 2.** Ozone scrubbers used in the tests

| Type | Description |
|------|-------------|
| Na$_2$S$_2$O$_3$ | PFA filter holder (Entegris, in-line filters with integral ferrule connection, Art.nr. 511-1 (1/8") or 511-2 (1/4")), O$_3$ filter (Pall, Pallflex T60A20) custom-made impregnation with sodium thiosulfate, baked out at 150°C for 2h after impregnation and a PTFE filter, pore size 20-30 micron (Entegris, Art.nr. 511-5, cut to 20mm diameterfilters, baked out at 100°C for 1 h). |
| Heated SS | 1/8" stainless steel tubing (grade 316), length 0.3 m, heated up to 120°C, cleaned by flushing with water, acetone and methanol before use |
| MnO$_2$ | Commercially available ozone scrubber which is used in Thermo Fisher Scientific ozone monitors. (Thermo Scientific PN 14697 ozone scrubber for 49i, Contrec AG, Dietikon) |
| Kl/Cu | 1/4" copper tubing inner surface coated with potassium iodide (KI), length 1 m |

**2.8 Tests for particulate filters**

Three types of particle filters were tested; Balston filters, PTFE membranes, SS screens (Table 3) to evaluate possible BVOC losses on filters commonly used in their measurements and analyses. Experiments were performed using a NPL gaseous standard cylinder containing α-pinene, β-pinene, limonene and toluene. Measurements with and without the particle filters were performed at a sampling flow rate of 0.100 L min$^{-1}$ and at 65-75 % relative humidity. Measured amount fractions were 275 ~1 nmol mol$^{-1}$. While other studies were conducted using a TD-GC-MS, these experiments were conducted using a thermal desorption unit connected to a TD-GC-FID. The sample was preconcentrated on a cold multi-sorbent trap (Tenax TA and glass beads) at -135 °C. Then the trap was heated at 120 °C and the sample passed along a second additional cold trap of Tenax TA at -55 °C where it was cooled and then heated at 180 °C. Finally, the sample reached a third trap where a flow of liquid nitrogen at -200 °C passed, then the sample was heated at 150 °C. The compounds were desorbed and injected into the GC for separation 280 and analysis with the FID. The separation was performed using a dual capillary column system of Al$_2$O$_3$/KCl (39 m x 0.32 mm x 5 μm) for C$_2$–C$_5$ hydrocarbons and CPSil 5CB (50 m x 0.32 mm x 1.2 μm) for C$_6$–C$_{10}$ hydrocarbons. The calibration of the TD-GC-FID was performed with a NPL calibration standard containing α-pinene, β-pinene, limonene and toluene at around 4 nmol mol$^{-1}$ for at least 5 replicates to reassure a good repeatability. The coefficients of variation were below 2%.

For the blanks, we conducted at least three injections of zero air injected into the TD-GC-FID to verify the background of the 285 zero air and subsequently the filters inserted to the inlet of the system to verify the background of each filter. Subsequently, the targeted compounds were inserted to the system and measurements with and without the filter were conducted for 3 to 4 repetitions. Supplement Fig. S2 shows the set-up of these experiments.





After considering the zeros and calculating the net normalized signal (NET = $M_{measured} - M_{zero}$, where M is the measured quantity) the relative differences (Eq. 6) between the value measured upstream the particle filter ($M_{WOF}$) and the value measured downstream the filter ($M_{WF}$) were calculated.

$$Rel.\,diff_o = \frac{M_{wf} - M_{wof}}{M_{wof}} \; x \; 100\%$$ (6)

**Table 3.** Type of particle filters used for the tests.

| Filter | Type/Supplier | Particle size (µm) | Comment |
|---|---|---|---|
| Balston new | Disposable filter unit/ Parker Balston | 2 | New filter |
| Balston aged | | 2 | New filter that has been used in earlier laboratory experiments |
| PTFE new | Hydrophopic PTFE membrane filter/ Sartorius | 5 | New filter |
| PTFE aged | | 5 | Filter charged indoors for 60h with a flow of 10 L $min^{-1}$ |
| Stainless steel new (F2) | Stainless steel filter support + filter element/ Swagelok | 0.5 | New filter |
| Stainless steel aged (F2) | | 0.5 | New filter that has been charged indoors for 41 h with a flow of 0.2 L $min^{-1}$ |
| Stainless steel used ($F_{used}$) | | 2 | Filter used in a field campaign |

**2.9 Laboratory intercomparison**

To evaluate the performance of the sorbent tubes under study, FMI and IMT Nord Europe conducted an interlaboratory comparison. For that purpose, IMT Nord Europe loaded 16 Tenax TA (mesh 60/80) and 16 Carbopack B (mesh 60/80) SS tubes with α-pinene, β-pinene and toluene using ReGaS2. The loading flow was 50 mL $min^{-1}$ for each tube. Half of the tubes of each sorbent material was loaded during 15 minutes and the other half during 30 minutes (see Supplement S1). Four sorbent tubes of each sorbent type were not loaded (i.e. blanks). The blanks were transported together with the sampling tubes to assess potential contamination during the transport. Each laboratory analyzed half of the tubes (including blanks).



FMI analyzed the tubes within two weeks after sampling using the TD-GC-MS methods described in this study and IMT Nord Europe performed the analysis immediately after sampling with a TD-(TurboMatrix 350, Perkin Elmer, United States) GC connected to a flame ionization detector (FID) (Clarus 680, Perkin Elmer) and a mass spectrometer (MS) (Clarus SQ8T, Perkin Elmer). The calibration of the TD-GC-FID/MS was performed with a NPL calibration standard containing α-pinene, β-pinene and toluene at around 4 nmol mol$^{-1}$. In order to cover the range of the amount fractions, we have established the linearity curve

with different sampling duration (15, 30 and 60 minutes), while keeping the same sampling flow (21 mL min$^{-1}$). The analytical method used by IMT Nord Europe has been described in detail by Debevec et al. (2021) and references therein.

## 3. Results and discussion

### 3.1 Storage stability


The National Institute for Occupational Health and Safety (NIOSH, US) recommends that if the average quantitative measurements of the samplers differ from the set analyzed on day 0 by more than 10 %, the method in question does not meet the sample storage stability criterion (Kennedy et al., 1996). Here one month storage stability in sorbent tubes was tested. For that purpose, both self-packed and commercial Tenax TA tubes and commercial Carbopack B tubes were used.

All studied compounds were relatively stable for one month in all tested Tenax TA tubes both at -20 oC and +22 oC (Fig. 2). Mean relative differences in the Tenax TA tubes kept at -20 oC and at +22 oC for one month compared to tubes analyzed immediately were -7 % to +13 % and -16 % to +16 %, respectively. These differences indicated that storage in a freezer may slightly improve storage stability. No significant differences were detected between SilcoNert 1000 and SS tubes or between self-packed and commercial tubes.Also, in earlier studies many monoterpenes have been found to be stable in Tenax TA or in

multibed sorbents containing Tenax TA for at least 14–62 days (Sunesson et al., 1999; Sheu et al., 2018; Helin et al., 2020). α-Pinene was stable even after 12 months of storage (Demichelis et al., 2009). However, Volden et al. (2005) had recovery of only 69 - 93 % for α-pinene and β-pinene in Tenax TA after storage of 7, 14 and 28 days at 5 ᵒC and 20 ᵒC. Earlier data on other BVOCs is more limited. Helin et al. (2020) studied several mono- and sesquiterpenes in Tenax TA – Carbopack B multibed sorbents and monoterpenes were recovered at 101±2 %, 93±5 % and 97±4 % after 5 days, 1 month and 2 months of

storage at 4 ᵒC respectively. Similarly, sesquiterpenes recovered on average at 104±2 %, 89±3 % and 94±5 % after 5 days, 1 month and 2 months of storage, respectively.

In Carbopack B tubes there were significant losses of most terpenes and only α-pinene, myrcene, p-cymene and 1,8-cineol had 100±10 % recovery at both temperatures (Fig. 2). On the other hand, recovery of camphene was clearly higher than expected (>170 %) which indicates that isomerization of terpenes is occurring in the tubes. It is also possible that some of the terpenes

(especially sesquiterpenes) are not fully desorbed from the tubes during the analysis. For aromatic hydrocarbons results were also good for the Carbopack B tubes, which was expected based on earlier studies (e.g. Hellén et al., 2002).





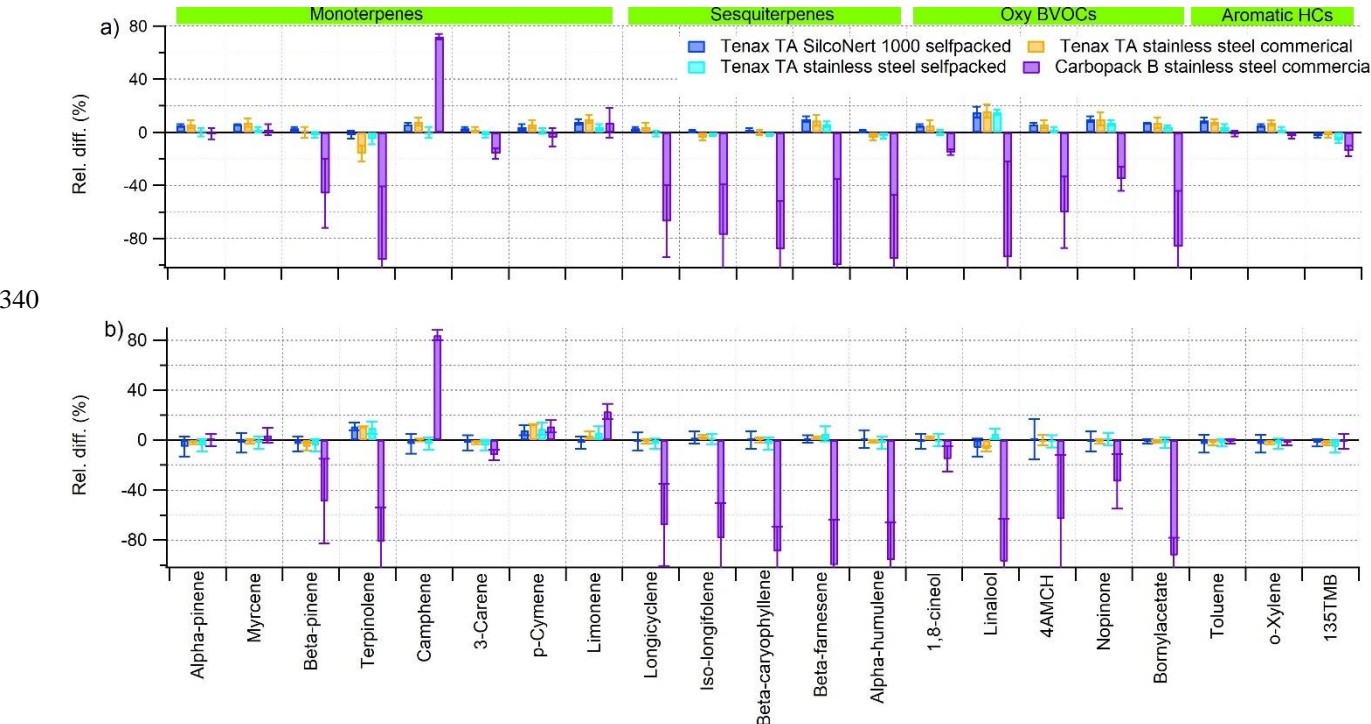

**Figure 2.** Relative difference of the sorbent tubes kept at a) +22 ºC and b) -20 ºC for one month compared to the tubes analyzed immediately. Error bars show the standard deviation (1σ) of the replicates.

### 3.2 Breakthrough volume

Camphene and α- and β-pinenes had the lowest breakthrough volumes of the studied compounds. Breakthrough volumes of camphene, α- and β-pinenes in the Tenax TA tubes were 4–16 L, 5–16 L and 8–26 L, respectively (Table 4). Some breakthrough was detected also for 3Δ-carene, limonene and 1,8-cineol with breakthrough volumes of 33–156 L, 38–158 L and 16–76 L, respectively. For self-packed tubes, volumes were lower than for commercial ones. However, tests with commercial tubes at 30 % humidity were done for the totally new tubes and this could explain the higher retention. In the tests under 70 % humidity conducted after the tests of 30 % humidity, any difference between commercial and self-packed tubes was not that clear. No breakthrough was detected for other BVOCs in Tenax TA tubes even with a sampling volume of 160 L. None of the studied compound was breaking through the Carbopack B tubes. Results for α- and β-pinenes from permeator tests were similar as with methanol injection (Fig. 3) and, therefore, methanol used as solvent was not expected to have significant effect on the breakthrough volumes.





**Table 4**. Breakthrough volumes (V_brkt, 5 %) of BVOCs measured by injecting methanol solutions of studied compounds into zero air.

| | Relative humidity (%) | Tenax TA commercial V_brkt (L) | Tenax TA self-packed V_brkt (L) | Carbopack B commercial V_brkt (L) |
|---|---|---|---|---|
| α-pinene | 30 | 16 | 5 | >100 |
| | 70 | 7 | 7 | >160 |
| β-pinene | 30 | 26 | 8 | >100 |
| | 70 | 11 | 9 | >160 |
| Camphene | 30 | 16 | 4 | >100 |
| | 70 | 7 | 7 | >160 |
| 3Δ-Carene | 30 | 156 | 33 | >100 |
| | 70 | 59 | 43 | >160 |
| Limonene | 30 | 158 | >45 | >100 |
| | 70 | >100 | 38 | >160 |
| 1,8-cineol | 30 | 76 | 16 | >100 |
| | 70 | 28 | 21 | >160 |
| Other BVOCs | 30 | >160 | >45 | >100 |
| | 70 | >100 | >45 | >160 |

Estimates of breakthrough volumes can also be found in the literature. Either artificially produced terpene rich air or ambient air has been used in the tests. Often two sorbent tubes were connected in series to follow the breakthrough from the front tube. Arnts (2010) found that Tenax TA tubes were able to trap α- and β-pinene efficiently at least at sampling volumes of 0.2–5 L. Gallego et al. (2010) found high breakthrough (48 % found in the back tube compared to front tube) of α-pinene in Tenax TA at a sampling volume of 10 L, while for the other studied terpene (limonene) breakthrough was much lower; only 2 and 6 % was found in the back tube with 40 and 60 L sampling volume, respectively. Sheu et al. (2018) found also problems in trapping α- and β-pinene with 10 L sampling volume into quartz wool-glass beads-Tenax TA multibed tubes, while for the limonene or the other studied >$C_9$ hydrocarbons, no breakthrough was detected. They also tested different sampling flows, but no effect of the flow on breakthrough volumes was observed. Veenas et al (2020) did not detect breakthrough (< 5 %) of α- and β-pinene with 4 L sampling volume while using the multibed sorbent containing Tenax TA and Carboxen 100. Helin et al. (2020) found that the breakthrough volume of all target terpenes was over 24 L with Tenax TA – Carbopack B multibed sorbent tubes.



Komenda et al. (2001) used Tenax TA-Carbotrap tubes and found that all studied terpenes are efficiently trapped at least up to
18 L. Helmig et al. (2004) tested Tenax TA, Tenax GR, Carbotrap, Carbotrap C, Unibeads and Glass Beads as sorbents for
sesquiterpenes including β-caryophyllene and found breakthrough from glass beads even with 2.1 L sampling volume.

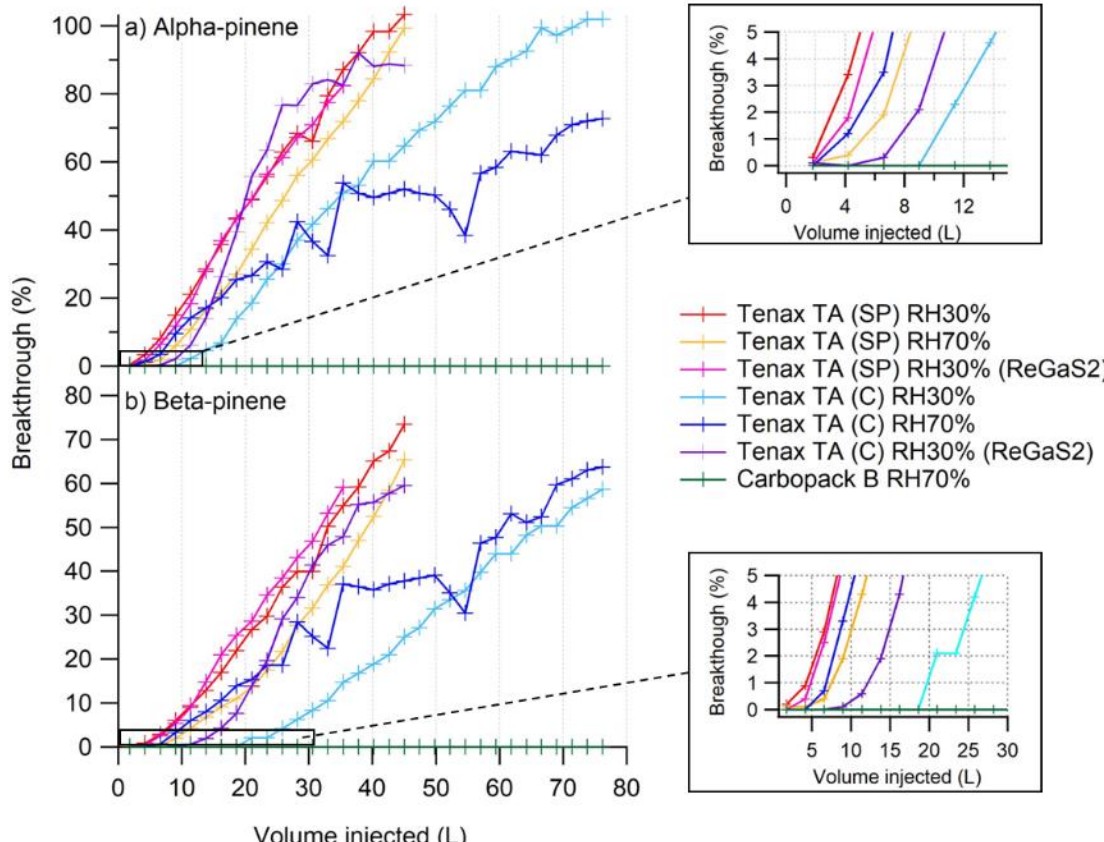

**Figure 3.** Breakthrough of a) α-pinene and b) β-pinene from Tenax TA and Carbopack B sorbent tubes (SP=self-packed tube,
C=commercial) at relative humidity of 30 % and 70 %. Monoterpenes were either injected into the zero air as methanol
solutions or generated by the portable generator (ReGaS2).

These earlier results are in accordance with the results presented in this study on α- and β-pinene having lower safe sampling
volumes in Tenax TA (4–7 L) than other $C_{10}$–$C_{15}$ BVOCs (>160 L) and breakthrough volumes in Carbopack B tubes being
very high (>160 L) for all studied compounds.

## 3.3 Sampling efficiency



From the comparison between the NPL gas standard and the methanol solution calibration methods, relative differences of
other compounds were within ±10 % criteria, but for limonene a bit greater (-20 to -15 %) difference to the expected amount
fraction of the NPL reference gas was detected (Fig. 4a). All three different sampling times showed similar recoveries.
However, with increased sampling time standard deviation between the replicates clearly decreased.

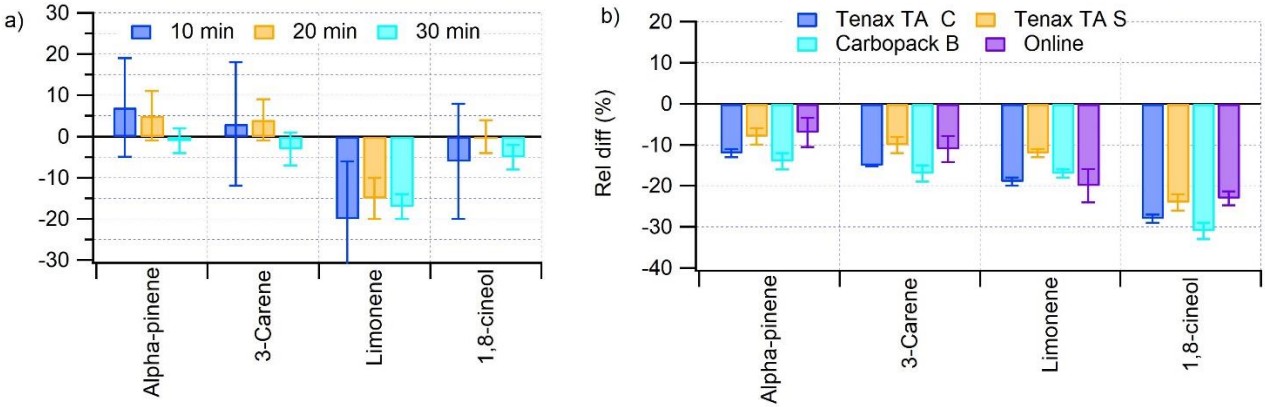


**Figure 4.** a) Recovery of NPL gas standard with different online mode sampling times. a) Recovery of the NPL gas standard
flushed into different tubes (C=commercial, S=self-packed) or analyzed with online mode of the instrument (online). Error
bars show the standard deviation between the  three replicate samples. For both tests the instrument was calibrated with
methanol solutions of BVOCs.


For most of the studied BVOCs, Tenax TA tubes showed good linearity with used sampling volumes (3, 6 and 12 L) and
sampled amounts (2–80 ng). Examples of the linearity fits can be found in the supplement Fig. S2. Decreased amount fractions
(~25 % lower than expected) were detected for α- and β-pinene and camphene with self-packed tubes with the highest sampling
volume of 12 L. This is in accordance with the breakthrough volume results where breakthrough volume of camphene, α- and
β-pinene was 4 L, 4 L and 8 L, respectively. In addition, it was not possible to verify the linearity of sesquiterpenes due to
instabilities in production of sesquiterpene rich air with the methanol injection system. Comparisons between the tubes and
online samples taken at the same time periods were still possible. Especially linalool and β-farnesene had higher yields in
Tenax TA sorbent tubes compared to online sampling (Fig. 5). In earlier studies, these compounds have been found to have
losses in the online sampling system (Helin et al., 2020).




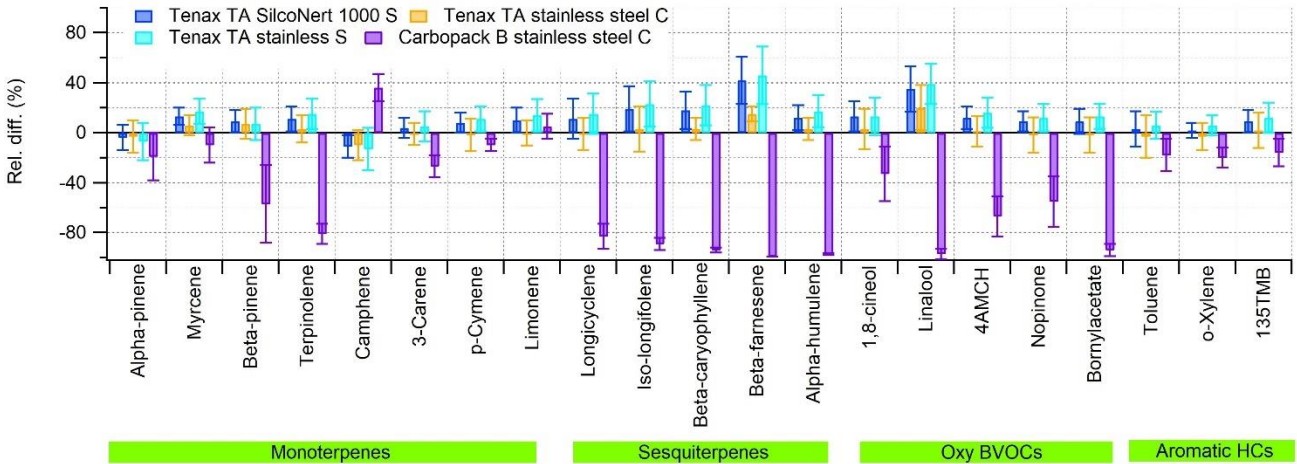

**Figure 5**. Relative difference (%) of the sorbent tubes (C=commercial, S=self-packed) compared to the online sampling results i.e. ratio of the difference between the sorbent tube and online sampling amount fractions to the online amount fractions. Relative humidity of the sampling air was ~70 %. Error bars show the standard deviation of the three replicate tubes taken with each sorbent sampling volumes (3, 6 and 12 L).

No clear difference between SilcoNert 1000 and SS tubes were detected in the linearity tests nor in comparison with online samples (Figs 5 and S2). The mean standard deviation of the six sets of parallel SilcoNert 1000 and SS tube samples was 4 %. Tests were conducted at relative humidity of 30 % and 70 %. Humidity of the sampling air did not influence the results.

In Carbopack B tubes most of the compounds had severe losses with all sampling volumes (Fig. 5). At the same time, amount fraction of some terpenes (e.g. camphene) increased. This indicates that some terpenes can be isomerized forming other terpenes in the Carbopack B tubes.

In the additional test using ReGas2, where sampled amounts of α-pinene varied from 14 ng to 44 ng, high linearity was observed. Standard deviation between the amount fractions measured with different sampling volumes was 1.2 %. For β-pinene amount fractions produced by ReGaS2 were out of the calibration range of the TD-GC-MS. Amounts of β-pinene collected varied from 40 ng to 103 ng while highest calibration point was 69 ng. Possibly due to exceeded calibration range, amount fractions measured with higher volumes 1.0 L (88 ng) and 1.2 L (103 ng) were 11 % and 16 % lower than amount fraction measured with only 0.4 L (40 ng), respectively.

The commercial Tenax TA tubes were also compared with online TD-GC-MS while sampling from ReGaS2. Both tubes and online samples were taken for 15 minutes with a flow of 40 mL min⁻¹. α-pinene, β-pinene and myrcene amount fractions in the tubes were 95 %, 100 % and 120 % of the online amount fractions, respectively.





In the sampling efficiency tests using NPL gas standard, a bit lower than expected amounts were measured especially for 1,8-
cineol with both sorbent tubes and the online TD-GC-MS (Fig. 4b). For sorbent tubes relative differences to the expected
amounts of α-pinene, 3Δ-carene, limonene and 1,8-cineol were -12 to -8 %, -17 to -10 %, -19 to -12 % and -31 to -24 %,
respectively. For online TD-GC-MS relative differences to the expected amounts of α-pinene, 3Δ-carene, limonene and 1,8-
cineol were -7 %, -11 %, -20 % and -23 %, respectively.

**3.4 Impact of tube material**

Results show that relative differences with different tube materials, were between -15 % and +20 % for most compounds, but
there were losses of some sesquiterpenes (β-caryophyllene, β-farnesene and α-humulene) and oxy BVOCs (linalool and 4-
AMCH) on SS tubes (Fig. 6). For these SS tubes slightly higher amounts of p-cymene and limonene were also detected. Earlier
studies by Helin et al. (2020) showed significant losses of β-farnesene, α-humulene, some oxygenated sesquiterpenes and
diterpenes as well, when using empty SS tubes. In their studies, recoveries were significantly improved by using glass tubes.
SilcoNert 1000, which was found in this study to be a suitable material too, was not tested in the earlier study. When tubes are
filled with sorbent material, the tube surface is much smaller and therefore sample has less contact with tube walls and losses
are expected to be smaller. This is supported by the other tests of our study, where losses of these compounds were not detected
(e.g. storage stability or sampling efficiency).


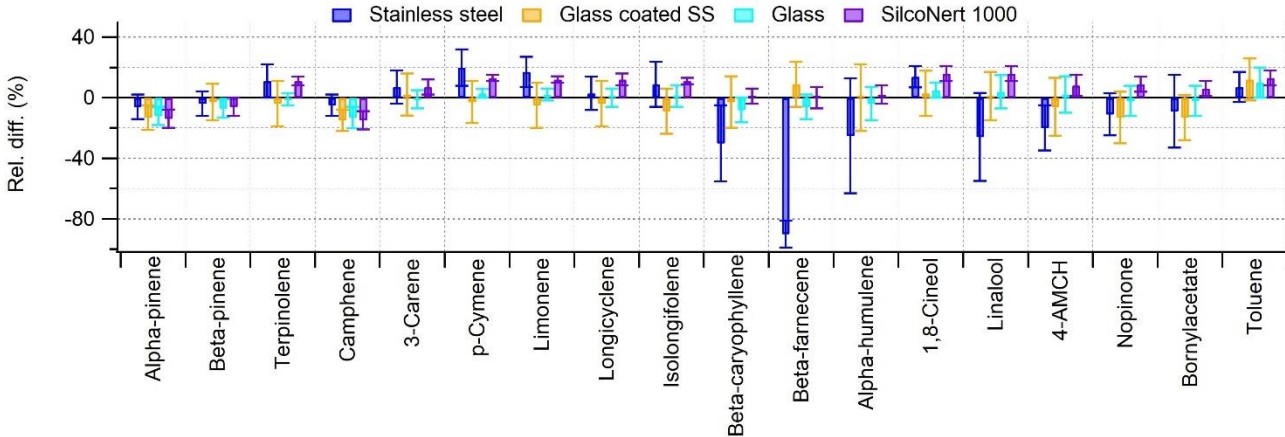

**Figure 6.** Relative difference of studied BVOCs flushed through the empty tubes (SS=stainless steel) compared to no tube
used situation (see Fig. 1 for the test set up).

**3.5 Recoveries from the ozone scrubbers**

Fig. 7 depicts an example of average amount fractions of the compounds, without (A) and with the ozone scrubber (B), with
(shaded) and without (unshaded) ozone. Much lower amount fractions were detected especially for myrcene, terpinolene and





β-caryophyllene without $O_3$ scrubbers when 40 nmol $mol^{-1}$ of $O_3$ was generated. This was expected since these BVOCs are most reactive with $O_3$.


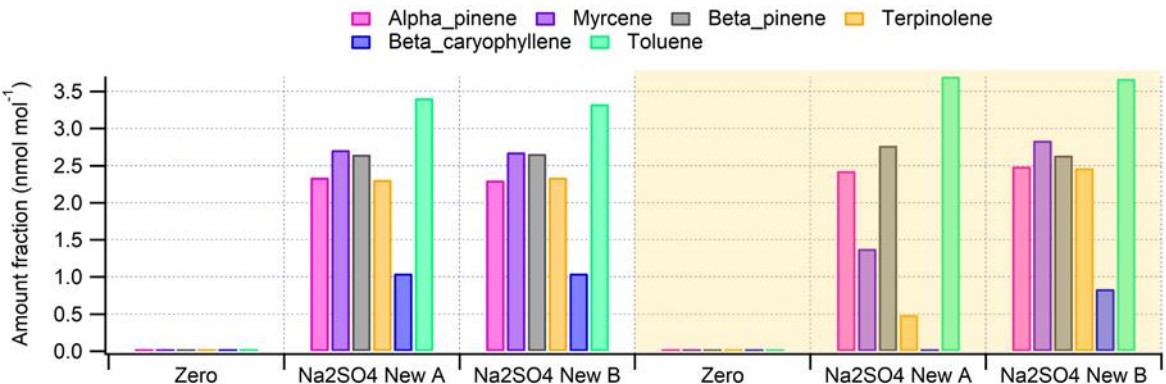

**Figure 7.** Example of amount fractions of studied compounds at each step of the experiment without (A) and with the ozone
scrubber (B). Shaded areas represent periods where ozone was generated with amount fraction of ~40 nmol $mol^{-1}$.

Relative difference of samples with and without $O_3$ scrubbers calculated by the Eq. 5 are presented for the tests without $O_3$ in
Table 5. Generally, new scrubbers were suitable for most of the BVOCs with relative difference below 5 % with some
exceptions like KI/Cu for β-farnesene, $Na_2S_2O_3$ for linalool, heated SS for linalool and bornylacetate (Table 5). However,
$MnO_2$-scrubbers were suffering on the losses of some monoterpenes, all sesquiterpenes and all oxy-BVOCs while recoveries
of aromatic hydrocarbons were good. $MnO_2$ scrubbers have been commonly used in monoterpene measurements (e.g. Hakola
et al., 2003, 2009), but also in earlier tests they have been found unsuitable for sesquiterpenes (Calogirou et al., 1996; Pollman
et al., 2005). In their studies Pollmann et al. (2005) recommended NO titration or sodium thiosulfate-impregnated filters as
suitable $O_3$ removal method.




**Table 5**. Relative difference between the samples taken with and without new and aged $O_3$ scrubbers for BVOCs and aromatic hydrocarbons and the flow through the scrubber during sampling.

| Scrubber type | Relative differences (%) | | | | | | | |
| --- | --- | --- | --- | --- | --- | --- | --- | --- |
| | $Na_2S_2O_3$ | | heated SS | | KI/Cu | | $MnO_2$ | |
| | at 80 mL min$^{-1}$ | | at 100 mL min$^{-1}$ | | at 1000 mL min$^{-1}$ | | at 100 mL min$^{-1}$ | |
| | new | aged | new | aged | new | aged | new | aged |
| Monoterpenes | | | | | | | | |
| α-pinene | -2 | 2 | 0 | 4 | -1 | -1 | 0 | 0 |
| myrcene | -1 | 1 | -2 | 4 | -1 | -1 | -6 | 0 |
| β-pinene | 0 | 2 | -2 | 2 | -1 | -1 | 0 | -7 |
| terpinolene | 1 | 3 | -2 | 6 | 0 | 3 | -27 | -33 |
| camphene | -2 | 1 | 0 | 3 | -3 | -1 | -1 | 6 |
| 3-carene | 1 | 4 | -2 | 5 | -1 | 1 | 1 | 4 |
| p-cymene | -1 | 2 | -2 | 4 | -1 | 0 | -1 | 4 |
| limonene | -1 | 2 | -2 | 6 | -2 | 0 | -1 | 9 |
| Sesquiterpenes | | | | | | | | |
| longicyclene | 1 | 6 | -2 | 7 | 1 | 3 | -54 | -45 |
| iso-longifolene | 1 | 0 | -2 | 8 | 3 | 3 | -66 | -58 |
| β-farnesene | -8 | -7 | -6 | 7 | -11 | -22 | -100 | -100 |
| β-caryophyllene | 0 | 4 | -4 | 7 | 5 | -5 | -98 | -96 |
| α-humulene | 3 | 7 | -6 | 10 | 6 | -6 | -100 | -100 |
| Oxy-BVOCs | | | | | | | | |
| 1,8-cineol | -2 | 3 | -2 | 2 | -3 | 1 | -51 | -46 |
| linalool | -12 | -3 | -10 | 1 | 2 | -18 | -100 | -100 |
| nopinone | -4 | 1 | -8 | 3 | 0 | 0 | -100 | -100 |
| bornylacetate | -6 | 1 | -16 | 6 | 2 | -1 | -100 | -100 |
| Aromatic HCs | | | | | | | | |
| toluene | -2 | 0 | -2 | 3 | -2 | -2 | 0 | 2 |
| o-xylene | -1 | 2 | -1 | 3 | -1 | 0 | 0 | 3 |
| 135TMB | -3 | 2 | -2 | 4 | 0 | -2 | 0 | 3 |



All $O_3$ scrubbers have limited ozone removal capacity and they must be changed and checked regularly depending on the season, the use and the ozone amount fractions. Since the $O_3$ removal efficiencies and BVOC recoveries are highly dependent on the used flows and sizes of the scrubbers (Hellén et al., 2012), results here are valid only for scrubber types and flows presented in Tables 2 and 5. For SS scrubber heating is also essential for both ozone removal and BVOC recovery.


### 3.5 Recoveries of terpenes from particle filters

The relative differences between the quantities measured without the particle filter and the quantities measured with the filter were calculated (Eq. 6). The results are depicted in Fig. 8.

Based on the results the application of a particle filter can affect the measured quantities of monoterpenes, mainly limonene,

in the range of ± 6 %. All type filters affect more limonene (<+7 %) than α-pinene and β-pinene (± 2 %). In addition, it is evident that a filter that has been used in a field campaign for longer time ($SS_{used}$ ($F_{used}$)) compared to a filter that has been charged shortly indoors ($SS_{aged}$ (F2)) can lead to higher differences. Thus, attention should be paid to the type of filters used for ambient measurements as well as to the maximum duration of use. During field campaigns, it is recommended to change the particle filter every 5 days to a week depending on the levels of particles and gases in the ambient air.


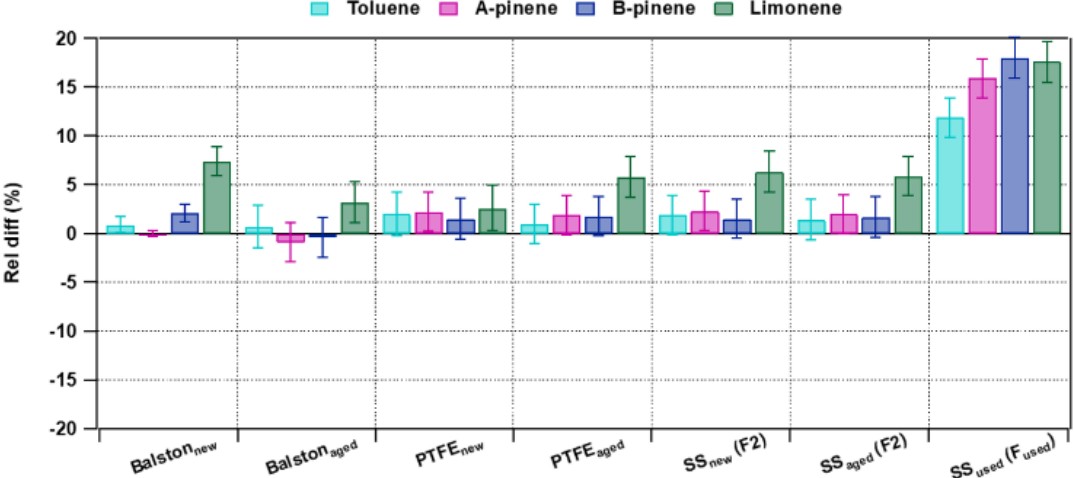

**Figure 8.** Relative difference between the measured quantities without and with the particle filters for terpenes and toluene. Error bars represent 1 σ.




## 3.6 Intercomparison of the sorbent tubes

For Tenax TA tubes, the relative standard deviations between the 4 replicate samples from each set up (15 or 30 minute samples for FMI and IMT) varied in the range of 0.4 – 1.4 %, 0.6 – 2.6 % and 0.4 – 4.6 % for α-pinene, β-pinene and toluene, respectively. The relative difference for α-pinene between measurements conducted for 15 minutes and the ones conducted for 30 minutes was 6.8 % and 6.1 % for FMI and IMT results, respectively. These deviations are at the same level as uncertainties (6.8 %, k=2). For β-pinene IMT 15-minute and 30-minute results were also in agreement (relative difference 5.1 % and uncertainty 6.2 %). Exception was β-pinene for FMI. The 30-min measurements of this compound led to an amount fraction out of the calibration range, which in turn led to higher relative difference (26 %) between the 15- and 30-minute samples of FMI.

For Carbopack B tubes relative standard deviation for the 4 replicate samples from each set up was higher than for Tenax TA tubes being 2 – 7 %, 10 – 28 % and 1 – 9 % for α-pinene, β-pinene and toluene, respectively. Also deviation between 15- and 30-minute samples was high for FMI samples especially for β-pinene (Fig. 9). The comparison between Tenax TA and Carbopack B tubes showed increased amount fractions of α-pinene and decreased amount fractions of β-pinene in Carbopack B tubes. In addition to that, both laboratories identified additional compounds (e.g. camphene and p-cymene) in the chromatograms obtained with Carbopack B tubes. This indication enhances the conclusion that in Carbopack B tubes losses of β-pinene can be attributed to the generation of other terpenes due to isomerization. Moreover, toluene showed less obvious differences between the two types of tubes.

When results of FMI and IMT were compared taking into account uncertainties, we had nice results for α-pinene with both sorbents (except 15-min loading with Carbopack B); and for β-pinene with Tenax TA (except 30- min loading, Fig. 9). More specifically for α-pinene the difference between laboratories was 23 % for Tenax TA for both sampling loadings, while for Carbopack B the relative differences were higher for the 15-min loading (13 %) compared to 30-min loading (1 %). For β-pinene the differences with Tenax TA were 6 % and 24 % for the 15-min and 30-min loading respectively. High difference for 30 min sample was due to exceedance of the calibration range of FMI. For β-pinene in Carbopack B the respective differences became 51 % and 1 %. Finally for toluene loaded in Tenax TA the differences between the two laboratories were 43 % and 20 % for 15-min and 30-min sampling loading respectively, while for Carbopack B were both in the order of 5 %. IMT results regarding toluene are reproducible between Carbopack B and Tenax TA. FMI had problems in the field blank measurements of Tenax TA tubes and high toluene could be originating from the blank accumulated during the storage and transport, which was not properly characterized. For other studied compounds no contamination was observed during transport. However, transportation may have had also impact on the compounds sampled on the FMI tubes and this could be the reason that sometimes differences between IMT and FMI were higher (23 % for α-pinene in Tenax TA) than the calculated uncertainty (7 % for both laboratories, Table 1, Supplement S2). Other possible reason is different calibration method; FMI calibrated with methanol solutions while IMT used NPL gas standard. Sampling efficiency tests in section 3.3 indicated that this can cause up to 12 % difference for α-pinene results.





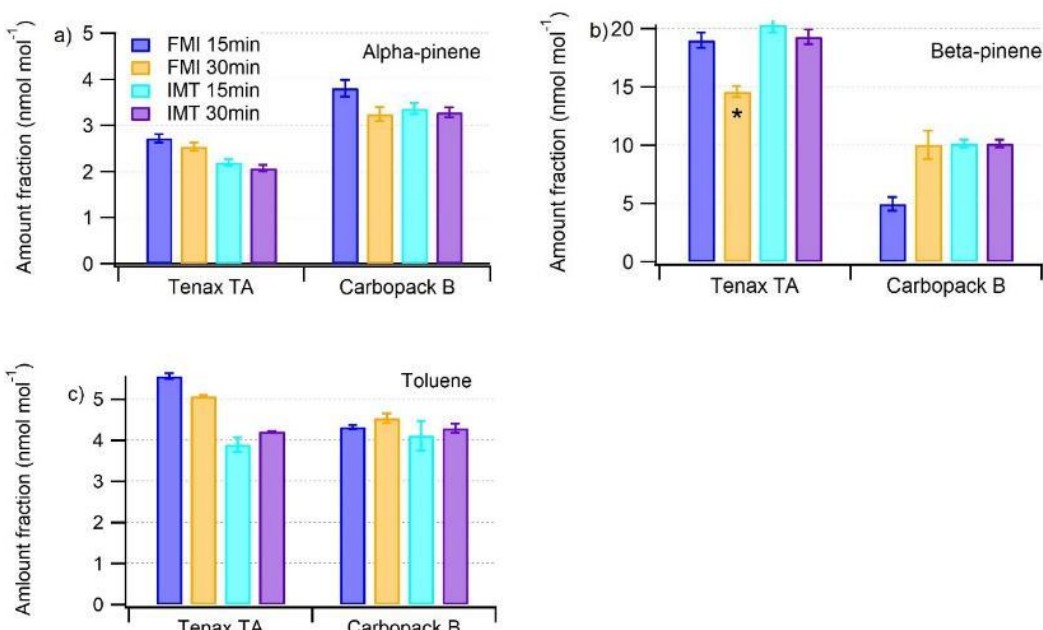

**Figure 9.** Amount fractions (in nmol/mol) of a) α-pinene, b) β-pinene and c) toluene in Tenax TA and Carbopack B tubes

sampled for 15 and 30 minutes and quantified by FMI and IMT Nord Europe laboratories. Error bars represent the average
expanded uncertainty (k=2, see Supplement S1) of the 4 measurements. The asterisk (*) indicates that β-pinene amount
fractions where out of the calibration range

## 4. Summary and conclusions

Sorbent tube tests showed that Tenax TA in stainless steel (SS)/SilcoNert 1000 tubes was more suitable for the measurements
of BVOCs than Carbopack B tubes, with which significant losses of several studied BVOCs were detected. At the same time,
when significant losses of several BVOCs were detected in Carbopack B tubes, amount fractions of some terpenes (e.g.
camphene) increased. This indicates that some terpenes can be isomerized forming other terpenes in the Carbopack B tubes.
Based on the earlier literature and results from this study, compounds were found to be stable in Tenax TA tubes for at least

one month at ±20 °C. Breakthrough tests indicated that α- and β-pinene have clearly lower breakthrough volumes in Tenax
TA (4–7 and 8–26 L, respectively) than other terpenes (>160 L) and breakthrough volumes in Carbopack B tubes are very
high (>160 L) for all studied BVOCs. SilcoNert 1000 or glass was shown to be best tube materials, but also SS tubes show
acceptable results due to low surface area of the tubes when filled with sorbents. Considering the cost of SilcoNert 1000 tubes
and fragility of glass tubes, SS tube may be the most convenient choice. Both $O_3$ scrubbers and particle filters may impact the

measured amount fractions of BVOCs. While KI/Cu, $Na_2S_2O_3$, heated SS scrubbers were found to be suitable for measurements



of studied BVOCs, MnO$_2$ scrubbers suffered on the losses of several BVOCs. Optimizing the sizes of scrubbers and used sampling flows is highly important for achieving good removal of O$_3$ and acceptable recoveries of BVOCs. All tested particle filters were shown to effect more on limonene than on α-pinene and β-pinene. Results also indicated that a filter that has been used in a field campaign for longer time can lead to higher differences compared to new filters or a filter that has been charged shortly indoors.

Tube sampling with Tenax TA and online sampling were in good agreement for most of the compounds. However, some losses of sesquiterpenes and linalool were detected in online mode compared to sorbent tube sampling. Sorbent tube calibration with methanol solutions of terpenes was within ±10 % when compared to the NPL gas standard for α- and β-pinene and 1,8-cineol, but for limonene 15 to 20 % lower than excepted amounts were found, when calibrated with the methanol solutions of terpenes. The study also included atmospheric oxidation products of monoterpenes (nopinone, 4-ACMH) and oxygenated monoterpenes (1,8-cineol, linalool). Results showed that Tenax TA sorbent tubes can be used also for studying them.

The laboratory intercomparison showed that in general, measured values by the two laboratories were in good agreement for terpenes measured with Tenax TA tubes. However, the comparison between Tenax TA and Carbopack B tubes showed increased amount fractions of α-pinene and decreased amount fractions of β-pinene in Carbopack B tubes. Furthermore, artefacts of additional compounds were observed in the chromatograms of Carbopack B tubes. This indication supports the conclusion that in Carbopack B tubes losses of β-pinene can be attributed to the generation of other terpenes due to isomerization. Moreover, toluene showed less obvious differences between the two types of tubes, which was expected based on earlier studies.

Even though online measurements of VOCs are becoming more and more common, the use of sorbent tubes is expected to continue due to greater spatial coverage and because no infrastructure is needed for sampling. Our findings showing that Tenax TA tubes are suitable as a sampling method of the offline measurement of most of the BVOCs under study confirms that sorbent tubes are still among the robust methods to get a wide range of speciated terpenes and other BVOCs.

For both offline and online measurements there is still a need for stable SI-traceable terpenes standards covering many terpenes at atmospheric amount fraction levels to overcome the matrix issue (methanol solution vs. gas standard) – e.g. only 4 terpenes were content in the gaseous SI-traceable standard used for this work.

*Acknowledgement.* We thank Albert Silvi (FMI) and Aku Helin (FMI) for conducting comparison tests with different tube materials, Thierry Léonardis (IMT) for conducting the intercomparison and for his technical support and Laurence Depelchin (IMT) for performing the particulate filter experiments for terpenes. Anja Claude and Stefan Reiman are thanked for providing sodium thiosulfate impregnated filters and MnO$_2$ nets as O$_3$-scrubbers.

*Financial support.* This research within the project 19ENV06 MetClimVOC has received funding from the European Metrology Programme for Innovation and Research (EMPIR) cofinanced by the Participating States and from the European Union's Horizon 2020 research and innovation programme.

*Competing interests.* The authors declare that they have no conflict of interest.



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
