# Peer review of "Measurements of atmospheric C10-C15 biogenic volatile organic compounds (BVOCs) with sorbent tubes"

_EGUsphere, 2023_

## Author Response (AR1)

Dear editor,

We have carefully considered all the reviewer comments on our manuscript 'Measurements of atmospheric C10-C15 biogenic volatile organic compounds (BVOCs) with sorbent tubes' and they have helped us improve the manuscript. We provide here below answers and mention the changes that have been made to address the referee's concerns. The referee's comments are in normal font while the replies are given in **bold**.

Yours sincerely,

Heidi Hellén

**Reviewer 1**

Hellen et al., in their paper "Measurements of C10-C15 biogenic volatile organic compounds (BVOCs) with sorbent tubes", investigated the performance of different configurations of adsorbent tubes in sampling BVOCs with respect to several parameters, including adsorbent type, tubing material, temperature, storage and ozone filtration techniques. They also cross-compared their results with a subset of measurements made at another research facility in a different institution (i.e. IMT Nord). The paper is well organized, properly structured and well written. I was also pleased to see that the authors thoroughly considered recent papers on this topic (as there are several!) and effectively identified areas of improvement to address through their paper. The topic of their research is quite relevant for offline measurement groups, especially those making ambient gas-phase measurements. Hence, I should think that this paper is very informative for relevant members of the scientific community, and also for those groups who may be considering using adsorbent tubes in future offline studies. However, I do have some concerns. I suggest publication of this work after these concerns are properly addressed:

My major concern is regarding the provided explanations for the observed differences in analyte recoveries between Tenax TA and Carbopack for different experimental conditions. I think that the explanations are a bit superficial and do not yet fully get to the scientific detail that will inform a confident decision making of future users of these techniques. For example, in line 354, the authors note that none of the studied compounds broke through the Carbopack B tubes, while some did in case of Tenax TA. At the same time, Figure 2 presents those analytes collected with Carbopack B tubes showed maximum losses over a 1-month storage period, while the losses were comparatively lower with Tenax TA. This seemed a bit counterintuitive to me. Yet, assuming that the results are all consistent, the authors should be able to provide some scientific explanation for why species would not break through an adsorbent material during sampling but show relatively higher losses from the same material when stored for a long period.

**-Carbopack B is a stronger sorbent for hydrocarbons than Tenax TA. Therefore, breakthrough volumes of Carbopack B are expected to be higher. However, storage stability does not only depend on the strength of the sorbent material, but also on the sampled compounds volatility and reactivity. In Carbopack B sorbent, some of these BVOCs seem to isomerize when interacting with the sorbent and therefore their stability in the sorbent is lower even without storage. As it was shown in the section 3.3, comparison of Carbopack B tubes with other tubes and online sampling showed that Carbopack B seemed to suffer on some losses already without long storage time. A comment on this was added to the manuscript into section 3.1 on lines 351-353 (new version of**

**the manuscript). While it is possible to keep sampling volumes low enough for Tenax TA samples, it is not possible to avoid deviations found for the Carbopack B tubes. Therefore, in the 'Summary and conclusions' -section we recommend Tenax TA as a sorbent for the studied BVOCs.**

In lines 333-334, isomerization of terpenes is suggested to be a source of high recovery of camphene from Carbopack tubes. Figure 2 shows that such high recovery of camphene was not observed in tubes other than Carbopack, suggesting material dependent behavior. Please explain this in more detail. Does Carbopack facilitate the isomerization process in some way?

**-We do not have detailed knowledge on the reactions occurring in the Carbopack B, but for all Carbopack B samples some of the monoterpenes (e.g. β-pinene) suffered losses while camphene was formed at the same time. This was especially clear in the intercomparison studies, where even if camphene was not flushed into the tubes, camphene peak was found from all Carbopack B tubes. Contrary to Carbopack B tubes, no camphene peak was observed in Tenax TA tubes when the same sample procedure was followed.**

In line 350, authors note that the breakthrough volumes were lower for self-packed tubes at 30% RH. Indeed, table 4 shows that the differences can be as high as 4-5x. This is an important information that warrants detailed explanation because self-packing can be impacted by biases pertaining to human activity (e.g. packing density achieved). Are the authors certain that this (or similar) difference would exist if another group replicated the self-made tubes? It is suggested that newer material in commercial tubes could explain the higher retention, but I would suggest expanding and providing a 2-liner scientific description of analyte retention in an adsorbent material. It seems that the retention tests with 70% RH were done in sequence after the 30% RH tests (lines 351-352), and then results were more comparable (Table 4). However, I imagine that the commercially packed adsorbent material would not age enough in sequential tests or does it? Also column 2 of table 4 shows a major drop in breakthrough volume when going from 30% to 70% for the commercial Tenax TA tubes. However, a drop of this magnitude is not observed in column 3 for self-packed tubes with the same material. Please provide a brief explanation for this and for the role of humidity in retention efficiency of different species.

**-We added a description of analyte retention in an adsorbent material and more discussion on the impact of humidity into the manuscript section 3.2 on lines 369-387 (new version of the manuscript): ' For the commercial tubes breakthrough volumes for RH 70% were lower than for RH 30 %. The impact of RH was not detected for self-packed tubes. Retention of the compounds on the sorbent is based on the competition of the analyte with other molecules in air. Even though Tenax TA is a hydrophobic sorbent at high humidity some water may still be adsorbed, or water may condense onto the tube impacting the sorption of other compounds. In earlier studies impact of RH on Tenax TA retention was not clear since often Tenax TA was studied in multibed sorbents. In their breath sample studies Wilkinson et al. (2020) found that high relative humidity during sampling generally reduces the ability of Tenax TA/Carbograph sorbent tubes to capture volatile compounds. However, since the impact was more pronounced for the less volatile compounds with higher breakthrough volumes, they speculated that impact is other than breakthrough. Ho et al. (2017) found significant impacts of RH on retention of $C_2$-$C_5$ aliphatic hydrocarbons in multisorbent (Tenax TA- Carbograph 1 TD- Carboxen 1003) tubes. The breakthrough volumes for $C_2$ aliphatic compounds were reduced by 13 – 22fold under 90 % RH and the main impact was**

expected to be through the less hydrophobic sorbent Carboxen 1003. For >$C_5$ compounds no breakthrough was detected. In the studies of Maceira et al. (2017) humidity problems were demonstrated with carbon-based tubes, while Tenax-based tubes did not display any influence. Our tests with commercial tubes at 30 % humidity were done for the totally new tubes and this could also explain the higher retention. In the tests under 70 % humidity conducted after the tests of 30 % humidity, any difference between commercial and self-packed tubes was not that clear. However, tests with ReGaS2 generator with commercial tubes at RH 30% conducted later show also bit higher breakthrough volumes, but not as high as for the first tests (Fig. 3), indicating some influence of RH. Increasing the temperature of the sorbent tube to 10 °C above the temperature of the air sample could reduce interferences of the humidity (Karbiwnyk et al. 2002).'

Figure 2: The error bars on the Carbopack tubes are quite broad irrespective of the storage temperature. Interestingly, in some cases, the errors are larger for tubes stored at -20 C. I would have expected reduced losses at -20 C relative to +20 C and thus smaller errors. It would be good if the authors could briefly explain this trend in section 3.1.

-We added to the manuscript into section 3.1 (lines 348-351 in the new version of the manuscript): 'Even though it would be expected that compounds are more stable at -20 °C relative to +20 °C, sometimes even larger deviations were observed in Carbopack B at -20 °C. Some deviations were detected also in the tubes analysed immediately possibly due to incomplete desorption or reactions on the sorbent surface.'

Lines 363-385: This reads like literature review. Further, since your results are consistent with previous work described in these lines (and stated as such in line 383), I suggest consolidating this paragraph more and stating more clearly how your work is novel in the aspect that is the focus of the discussion in this section.

-We modified section 3.2 (lines 398-408 in the new version of the manuscript) and added a comment that in most earlier studies only a few monoterpenes (α- and β-pinene) have been studied while here several different BVOCs were studied. Modified version: 'Estimates of breakthrough volumes can be found in the literature, but most earlier studies included only α-pinene, β-pinene and/or limonene (Arnts, 2010; Gallego et. al., 2010; Sheu et al., 2018) while in our study several BVOCs were tested. These earlier results are in accordance with our findings on α- and β-pinene having lower safe sampling volumes in Tenax TA (4–7 L) than limonene (≥ 38 L). In addition to this, our study found that other $C_{10}$–$C_{15}$ BVOCs have also higher safe sampling volumes except camphene, which was breaking through at same volumes as α- and β-pinene. In studies on Tenax TA in multibed tubes together with stronger carbon-based sorbents (Veenas et al. 2020, Helin et al. 2020, Komenda et al, 2001), much higher breakthrough volumes (even >24 L) were detected even for α- and β-pinene. Also, this is in accordance with our results on breakthrough volumes in Carbopack B tubes being very high (>160 L) for all studied compounds. Due to lower amount of each sorbent in multibed tubes, especially α- and β-pinene may be breaking through the Tenax TA into the carbon-based sorbent at very low sampling volumes. Behaviour of the compounds in these carbon-sorbents should be also considered as shown by losses and possible isomerization of some terpenes (e.g. β-pinene) in Carbopack B in this study.'

Figure 6: Why do you see up to 20% enhancement in signal when an empty tube is used ahead of the sorbent tube (Figure 1)? I would have expected either the same result or some upstream losses to the surface of the empty tube making the relative difference either negative or zero. Interestingly, the error bars on some compounds with positive differences are bounded in the positive domain (i.e. lower error bound > 0), especially for Carbopack-based measurements. Does this mean all replicates consistently showed enhancement in signal when an empty tube was used upstream? If so, why? All these aspects should be explained in section 3.4.

**-This test set up had higher than average uncertainty, which was shown in table 1. This was caused by the fluctuations in sampling flows and higher than normal drifts in MS signal. To explain this we modified the first paragraph of section 3.4 (lines 469-473 in new version of the manuscript): 'The results show that the relative differences with different tube materials, were between -15 % and +20 % for most compounds. This is expected to be within uncertainty of this test set up. Higher than average uncertainty (Table 1) was caused by the fluctuations in sampling flows and drifts of the MS response. However, there were clear losses of some sesquiterpenes (β-caryophyllene, β-farnesene and α-humulene) and oxygenated BVOCs (linalool and 4-AMCH) on SS tubes (Fig. 6). For these SS tubes slightly higher amounts of p-cymene and limonene were also detected.'**

Lines 521-522: Please explain why the probability of isomerization of beta-pinene would depend on adsorbent material. Similarly, explain the opposite trend for alpha-pinene. This will be useful for future users.

**-In Tenax TA no additional terpene peaks were detected. It only happened in Carbopack B. We added a comment on this into section 3.6 on lines 554-555 (the new version of the manuscript).**

Lines 534-537: The rationale provided here about the impact of transportation sounds vague, particularly when explaining the 23% difference for alpha-pinene. I would imagine that the tubes were transported in a controlled environment. Please provide more detail regarding the role of transportation in causing this discrepancy. Again, this is important for other users since being able to sample in remote locations is a major advantage of offline techniques and transportation is a key element between sampling and eventual laboratory analysis of a collected sample.

**-Samples were transported through regular post without environment controlling (e.g. cooling). We added a comment on differences of the laboratories to the manuscript into section 3.6 and lines 570-575 in the new version of the manuscript: 'Some differences between IMT and FMI were higher (23 % for α-pinene in Tenax TA) than the calculated uncertainty (7 % for both laboratories, Table 1, Supplement S2). A possible reason is the different calibration method used and their associated uncertainties; FMI calibrated their TD-GC-MS with methanol solutions while IMT used a traceable NPL gas standard to calibrate their TD-GC-FID. Comparisons of liquid methanol standard and NPL gas standard in section 3.3 indicated that this can cause up to 12 % difference for α-pinene results. However, to a lesser extent, transportation through regular post without cooling may have had also impact on the compounds sampled on the FMI tubes.'**

Minor points:

Line 100: Suggest renaming "Experimental" to "Experimental techniques" or something similar. "Experimental" on its own is incomplete.

**-corrected**

Table 1: Font correction required.

**-corrected**

Lines 320-321: The symbol for degrees should be in superscript. Please check at other places.

**-corrected**

Line 498-499: Please provide citation or supporting data.

**-this sentence has been removed as suggested by the other reviewer**

Line 524: Avoid adjectives – "nice".

**-the sentence was reformulated (Section 3.6, lines 558-560 in the new version of the mansucript): 'When results of FMI and IMT were compared taking into account uncertainties, the results for α-pinene with both sorbents (except 15-min loading with Carbopack B); and for β-pinene with Tenax TA (except 30- min loading, Fig. 9) were in agreement.'**

**Reviewer 2:**

Overall assessment:

Despite the increasing use of online measurement techniques (e.g., PTRMS, online GC), sorbent tubes remain an important tool used by the scientific community to quantify BVOCs (isoprene, monoterpenes, sesquiterpenes, etc.). The authors present results on breakthrough volumes and storage stability for different VOCs and with different sorbent and tube materials. These results are important and should be made available to the wider scientific community. The manuscript is well-written, and the methodology and data interpretation conducted by the authors appear sound. The results are clearly presented in figures/tables and support the conclusions of the study. I would suggest several revisions to further improve the manuscript (see below).

General comment:

Why not also evaluate the measurements of isoprene using sorbent tubes, which is by far the most abundant BVOC (accounting for about 50-70% of global BVOC emissions)?

**-Focus of this project was less volatile biogenic compounds, which have not been studied as much as isoprene. Isoprene measurements with online and offline techniques are better established. Terpene measurements including oxygenated BVOC, are more challenging and up to now few online and offline measurement studies focused on studying this large panel of species. Moreover, it is well-known that the tested Tenax TA tubes are not suitable for isoprene measurement.**

Line-by-line comments:

L16: Replace "oxy BVOCs" with "oxygenated BVOCs" for greater clarity.

**-corrected**

L17: "bornylacetate": there should be a space between "bornyl" and "acetate". Please change this throughout the manuscript as well.

**-corrected**

L24: What do you mean by ±20 °C? Do you mean from -20 °C to +20 °C? Also see Line 555.

**-this was corrected to 'at -20 °C and at +20 °C'.**

L50: Maybe also mention PTR-Q-MS (quadrupole mass spectrometer)?

**-we changed the sentence into 'Proton transfer reaction mass spectrometers (PTR-MSs) both quadrupoles and time-of-flights (TOFs)) have been used...'**

L68: Change "Most commonly sorbent," to "Most common sorbent"

**-corrected**

L104: delta-3-carene is written incorrectly.

**-corrected**

L107: What do you mean by "Each compound was weighed in 500 mL of methanol"? Please revise this sentence to make it clearer.

**-For clarification the sentence was changed to 'For producing standard solutions ~30 mg of each BVOC was added to the  500 mL of methanol, which was further diluted into six different amount-of-substance fractions (a.k.a. amount fractions).'**

L114: The word "online" is repeated twice.

**-corrected**

L124: Please use proper superscripts for the degree symbol in 320°C and for the units for flow rate.

**-corrected**

Table 1: What are the LOQs for Carbopack B tubes?

**-LOQs for Carbopack B tubes were added to the table 1.**

L153: Do you mean "dual stage regulator"?

**-yes, this was corrected.**

L155: How did you sample the sorbent tubes directly from the outlet of the gas cylinder valve? Did you use a pressure or flow regulator to limit the flow going through the sorbent tube?

**- Clarification for this was added to the section 2.2 (lines 160-162 in the new version of the manuscript): 'Sorbent tubes were sampled from the outlet of the gas cylinder valve using a T-**

connector and a pump (N 86 Laboport, KNF) with the sampling flow of ~100 ml min$^{-1}$. The main standard gas flow was kept at ~200-500 ml min$^{-1}$ during sampling.'

L214: How did you control the relative humidity?

**-We added a clarification on this to the section 2.5 (lines 234-236 in the new version of the manuscript): 'The air was humidified for desired level by bubbling a fraction of air through the ultrapure water (Milli-Q Gradient, Molsheim, France). The RH was measured with a Vaisala HMI 33 device (probe HMP 35, Vaisala, Helsinki, Finland).'**

L215: This sentence in unclear: "Since no blank was detected in blank sorbent tube tests, zero was used as one point."

**-due to unclarity the sentence was removed**

Figure 1: What is the model/manufacturer of the sampling pump used to pull the air stream through the sorbent tubes? This should be mentioned somewhere in the methods section.

**-this was added to section 2.1**

Table 2: "diameterfilters" should be two words.

**-corrected**

L365-376: *Just a suggestion* It might be helpful to summarize this information in a table to make it easier for the reader.

**-as suggested by the reviewer 1, we condensed this section**

L405: Is there a different way to generate sesquiterpenes of varying amount fractions without using the methanol injection system? It would be nice if you could also perform the linearity tests on sequiterpenes.

**-There are no gas standards or reliable permeation tube methods available for sesquiterpenes. However, with our methanol solution injections directly into sorbent tubes under nitrogen flow, we have found similar calibration curves as for the monoterpenes, which gives some confidence for their calibration even though traceability cannot be achieved.**

L422: Do you have an explanation/guess for why most of the compounds had severe losses in Carbopack B tubes? Is Carbopack B not a suitable sorbent for analyzing mono- and sesquiterpenes? This should be addressed somewhere in the manuscript.

**-we added into Summary and conclusions -part (lines 588-589 in the new version of the manuscript) sentence 'As shown by the losses of sesquiterpenes, oxygenated BVOCs and some monoterpenes during storage, in comparisons with online sampling and in laboratory intercomparisons, Carbopack B is unsuitable sorbent for the studied BVOCs.'**

L495: Can you provide an explanation/guess as to why limonene is affected more than a-pinene and b-pinene?

**-Higher reactivity of limonene could be a reason. This was added to the manuscript section 3.5 (lines 527-528 in the new version of the manuscript).**

Figure 8 legend: Use Greek letters for a-pinene and b-pinene, or write in full, e.g., alpha-pinene, beta-pinene.

**-The legend of Figure 8 is corrected:**

[Figure]

L547: Replace "where out of the calibration range" with "were out of the calibration range".

**-corrected**

L561: Maybe state which specific BVOCs suffered the greatest losses with MnO2 scrubbers.

**-this was added**

L585: "were content": Do you mean "were contained"?

**-yes, this was corrected**

**Reviewer 3:**

This manuscript describes a series of experiments characterizing solid adsorbent tubes for the sampling and thermal desorption of biogenic volatile organic compounds. Overall, this is a nice piece of work and a well conducted study.

What I wonder, though, is what the groundbreaking new findings of this work are? Most of what is described in this paper has been done in prior studies and been published in one way or another. The findings that are presented in this manuscript mostly confirm the prior literature. Of course, there is some value in this, however, the paper could be strengthened if the authors were to explicitly emphasize what the new findings from this work are and how they advance the science in this field.

**-Even though there are some studies on suitability of sorbent tubes for BVOCs, usually only a few most common terpenes (e.g. alpha and beta-pinene) are studied and very little is known on the suitability of the sorbent tube sampling for oxygenated BVOCs. Performance of the tube sampling method may vary a lot even for different monoterpenes. We tried to clarify this in the**

introduction. Ozone traps and tube materials have been studied, but not for this range of BVOCs, neither particle filters were thoroughly addressed for BVOCs in the literature to the best of our knowledge. During recent years it has become clear that more BVOC measurements (in addition to isoprene) are needed for air quality monitoring in rural areas in a climate change context, and in urban areas, where terpenes can be markers of some anthropogenic sources. Therefore cost-effective, simple and reliable sampling methods are needed. In addition to sorbent tubes, many of these results can be used also for choosing the most suitable set-up, cold trap and optimizing sampling times and flow for online sampling of BVOCs (breakthrough, selection of ozone scrubbers, tube materials, particle filters…). Carbon-based sorbents (e.g. Carbopack B) are still sometimes used for measurements of terpenes and quite often in multibed tubes. This study clearly shows that Carbopack B is not suitable for these compounds and even when using multibed tubes one should take care that these BVOCs are not breaking through the Tenax TA into the carbon sorbent.

There are a few places where the language is a bit inaccurate. I encourage the authors to carefully and critically review the entire manuscript text. A few suggested corrections are presented below.

**-The new version of the manuscript was sent to language check**

Specific comments:

Line 1: Measurements of atmospheric C10-C15 biogenic …..

**-title changed as suggested by the reviewer**

13: …typically present at sub parts per billion mole fractions in

**-this was added**

15: … simple sampling apparatus is needed for sample collection.

**-this was added**

24/25: It would be better to report breakthrough volumes normalized to the amount of adsorbent.

**-Since breakthrough volumes may also depend on the shape of the sampler and since all the other tests are based also on this certain type of sampler, we did not normalize these to the amount of adsorbent. It is easier for the users of the tubes to have these as volumes. We added a comment that these volumes are for the used type of tubes. Also in EN 14662-1 standard they use volume/tube.**

28: Be more specific than 'greater impact'.

**-We added relative difference values**

34: The global atmospheric burden of biogenic volatile …..is ~ 10 times

**-this was corrected as suggested**

56: In my understanding of terminology, I would not label sorbent sampling-chromatography methods as 'in-situ'.

**-That is true, sorbent tube sampling with chromatographs is not in-situ, but these chromatographs are often also used in-situ. Then the chromatograph is at the sampling site and the sample is taken directly into the cold trap of the thermal desorption unit of the chromatograph. Those are generally called as in-situ/on line measurements. We tried to clarify this in the manuscript in section 2.2 (lines 162-163 in the new version of the manuscript).**

85: There are quite a number of options for selective removal of ozone from the sample air, see for instance [*Helmig*, 1997]. Why and how did the authors select and decide to investigate the methods described in their study?

**-these are commonly used at European air quality monitoring stations and are suitable for sorbent tube sampling.**

107: What do you mean by "Each compound was weighed in 500 mL of methanol"?

**-Clarification on this was added to the manuscript into section 2.1 (lines 114-115 in the new version of the manuscript): 'For producing standard solutions ~30 mg of each BVOC was added to the 500 mL of methanol, which was further diluted into six different amount-of-substance fractions (a.k.a. amount fractions).'**

110: How high was the air flow rate?

**-Flow rate varied between 1 to 2 L/min depending on the test. This was added to the section 2.1.**

114: …online online ….

**-corrected**

125: This seems like a very short tube conditioning time?

**-Freshly packed sorbent tubes were conditioned for longer periods, typically hours. This was re-conditioning time after every use. We added this remark into the manuscript.**

153: Please give the standard preparation date and please specify the regulator that was used.

**-these were added to the manuscript section 2.2**

154: Please give flow rate and sampling length time.

**-They were added to the manuscript section 2.2**

159: How was the noise level determined if there were no peaks detected in blank runs?

**-There is always background noise in the chromatograms, but no peaks deviating from the noise were detected in the blanks. Signal-to-noise ratio was calculated from the sample and calibration runs by integrating the noise next to the actual peak.**

167: What makes you believe that the storage behavior of diluted methane solutions injected onto the cartridges is the same as for collected gas samples (without a large excess of methanol)?

**-Similar as to the calibration tubes, the methanol was flushed away before the storage and analyses. Excess methanol would impact the analyses even without storage. We added a comment on this to the manuscript section 2.3 (lines 196-197 in the new version of the ,amuscript).**

184: Please provide more detail on the permeation device.

-permeation device was described in more details in the supplement S1 and we added a reference to the supplement into the manuscript on line 213.

250:  ….sodium thiosulfate ( …..

**-corrected**

251:  ….manganese dioxide …

**-corrected**

252:  These flow rates vary by more than a factor of ten for the different scrubbers that were tested.  How do you know that the comparison results are due to the materials tested and not from the different flow rates?

**-It is true that these results depend on the flow rates. In addition to possible losses of VOCs, the flow rates depend on the ozone removal capacity of the scrubbers. While for the recoveries of VOCs high flows would be beneficial, with too high flow ozone is not removed efficiently anymore or ozone removal capacity of the scrubber remains efficient during very short time period (ozone removal capacities have been studied in earlier studies e.g. Hellén et al. 2012). The flows used for different scrubbers have been optimized based on this and are representing the flows used in the field. We have commented this in the results section 3.5 (lines 560-562 in the new version of the manuscript).**

Table 2:  Please explain the impregnation procedure.

**-added to the manuscript**

Table 2:  Please explain the coating procedure with KI.

**-The coating procedure is as follows:**

**"The potassium iodide ozone trap was home-made prepared by using a one meter copper tube (1/4 inch diameter) coated internally with a semi-saturated solution of potassium iodide (KI) consisting of 12.75 g of KI dissolved in 20 mL of demineralized water, for 3 hours. Then, the tube is emptied from the remaining solution, before being dried under a constant flow of zero air (0.75 L min$^{-1}$) for at least 3 hours." Short version of this was added to the Table 2**

275-282:  It would fit better to include the description of this instrument in the methods section.  Please also provide the details of the focusing traps.

**-This part was moved to section 2.2.**

**Here are additional details regarding the three modules of the trap summarized in the table below:**

TABLE**XX: D**ETAILED DESCRIPTION OF THE TRAPS PRESENT ON THE PRECONCENTRATOR.

| Trap | Trap characteristics | Goal | Principle | Parameters and comments |
|------|----------------------|------|-----------|-------------------------|

| | | | | |
|---|---|---|---|---|
| M1 | glass beads + Tenax TA | $H_2O$ elimination | (i) cooling = total trap (VOC+$H_2O$+$CO_2$) | (i) T = -135 °C |
| | | | (ii) preheating = equilibrium between condensed and gas phases | (ii) T = +10 °C |
| | | | (iii) thermodesorption: transfer of VOCs+$CO_2$ to M2 | (iii) T = +120°C (heating by electrical resistance) |
| M2 | adsorbent: Tenax TA | $CO_2$ elimination | (i) cooling = VOC trap +$CO_2$ (ii) thermodesorption: transfer of VOCs to M3 | (i) T = -55 °C (ii) T = +180 °C (heating by electrical resistance) |
| M3 | megabore cryofocusing tube | cryofocusing of the sample for flash injection | (i) thermodesorption: transfer of VOCs to GC | (i) initial T = -200 °C (with liquid $N_2$) T=+150°C (heating with hot air by passing via aluminum plate) |

**To clarify this we modified the description of the instrument now in section 2.2 lines 181-186 in the version of the manuscript as follows: 'Samples entered the instrument through a preconcentrator provided with three traps. In a first step of the preconcentration, samples passed through a cold multi-sorbent trap (Tenax TA and glass beads) at -135 °C to eliminate the water in the sample. Then the trap was heated to 120 °C and the sample passed along a second additional cold trap of Tenax TA at -55 °C where it was cooled and then heated to 180 °C to remove the $CO_2$ present in the samples. Finally, the sample reached a third trap where a flow of liquid nitrogen at -200 °C passed, then the sample was heated to 150 °C.'**

Table 3: Please provide the part numbers of the filter materials that were tested.

**The part number of each type of PM filter was added to Table 3 into the manuscript.**

**Table 3. Type of particle filters used for the tests.**

| Filter | Type/Supplier | Particle size (µm) | Comment |
|---|---|---|---|
| **Balston new** **Balston aged** | **Disposable filter unit/** **Parker Balston** **Part number 9933-05-BQ** | **2** **2** | **New filter** **New filter that has been used in earlier laboratory experiments** |
| **PTFE new** **PTFE aged** | **Hydrophopic PTFE membrane filter/** **Sartorius** **Part number 11842** | **5** **5** | **New filter** **Filter charged indoors for 60h with a flow of 10 L min$^{-1}$** |
| **Stainless steel new (F2)** **Stainless steel aged (F2)** | **Stainless steel filter support + filter element/** **Swagelok** **Part number SS-4F-K4-05** | **0.5** **0.5** | **New filter** **New filter that has been charged indoors for 41 h with a flow of 0.2 L min$^{-1}$** |

| | | | |
|---|---|---|---|
| Stainless steel used (F$_{used}$) | Part number SS-4F-K4-2 | 2 | Filter used in a field campaign |

307: Provide all detail on the NPL standard (including preparation date).

**-The date of issue and comment on the standard composition was added to the manuscript section 2.9 on lines 322-324 (new version of the manuscript): 'The calibration of the TD-GC-FID/MS was performed with a NPL (UK, date of issue 20th March 2020) calibration standard containing NMHCs from C$_2$ to C$_9$ (including toluene) and 3 monoterpenes (α-pinene, β-pinene, limonene) at around 4 nmol mol$^{-1}$.'**

**The composition of the NPL standard is as follows:**

**AMOUNT FRACTIONS:**

| Component | Amount fraction / (nmol/mol) | | Component | Amount fraction / (nmol/mol) | |
|---|---|---|---|---|---|
| ethane | 4.44 ± | 0.09 | isoprene | 4.59 ± | 0.12 |
| ethene | 4.35 ± | 0.11 | *n*-heptane | 4.61 ± | 0.10 |
| propane | 4.38 ± | 0.09 | benzene | 4.35 ± | 0.09 |
| propene | 4.36 ± | 0.09 | 2,2,4-trimethylpentane | 4.33 ± | 0.09 |
| 2-methylpropane | 4.47 ± | 0.12 | *n*-octane | 4.34 ± | 0.09 |
| *n*-butane | 4.43 ± | 0.09 | toluene | 4.23 ± | 0.11 |
| ethyne | 4.59 ± | 0.28 | ethylbenzene | 4.58 ± | 0.12 |
| *trans*-but-2-ene | 4.44 ± | 0.09 | *m*-xylene + *p*-xylene | 8.91 ± | 0.23 |
| but-1-ene | 4.42 ± | 0.09 | *o*-xylene | 4.38 ± | 0.11 |
| *cis*-but-2-ene | 4.43 ± | 0.09 | 1,3,5-trimethylbenzene | 4.18 ± | 0.11 |
| 2-methylbutane | 4.37 ± | 0.09 | 1,2,4-trimethylbenzene | 4.42 ± | 0.12 |
| *n*-pentane | 4.39 ± | 0.09 | 1,2,3-trimethylbenzene | 4.20 ± | 0.11 |
| 1,3-butadiene | 4.48 ± | 0.09 | (+/-)-α-pinene* | 4.62 ± | 0.19 |
| *trans*-pent-2-ene | 4.41 ± | 0.09 | (+/-)-β-pinene* | 4.50 ± | 0.14 |
| pent-1-ene | 4.48 ± | 0.09 | limonene* | 4.51 ± | 0.14 |
| 2-methylpentane | 4.60 ± | 0.10 | nitrogen | balance | |
| *n*-hexane | 4.60 ± | 0.10 | | | |

*components outside UKAS and CIPM MRA scope

The reported expanded uncertainties are based on standard uncertainties multiplied by a coverage factor $k = 2$, providing a coverage probability of approximately 95 %. The uncertainty evaluation has been carried out in accordance with UKAS requirements.

346: I question that breakthrough test results from samples that have such a high excess of methanol are applicable to gas standard samples.

**-To ensure that the methanol was not affecting the results, we measured the breakthrough volume also with the permeation unit. As the gained results do not differ significantly, we assume the methanol plays a minor role (if any).**

385: Please discuss the effect of humidity in more depth. There definitely seem to be some differences. And 70% RH isn't really all that high. For enclosure experiments with BVOC sampling one will often have sustained higher RH levels.

**-There were differences for the commercial Tenax TA tubes, but not for self-packed. However, tests with commercial tubes at 30 % humidity were done for the totally new tubes and this could**

explain the higher retention. We added to the manuscript section 3.2 (lines 369-385 in the new version of the manuscript): '. For the commercial tubes breakthrough volumes for RH 70% were lower than for RH 30 %. The impact of RH was not detected for the self-packed tubes. The retention of the compounds on the sorbent is based on the competition of the analyte with other molecules in air. Even though Tenax TA is a hydrophobic sorbent at high humidity some water may still be adsorbed, or water may condense onto the tube impacting the sorption of other compounds. In earlier studies impact of RH on Tenax TA retention was not clear since often Tenax TA was studied in multibed sorbents. In their breath sample studies Wilkinson et al. (2020) found that high relative humidity during sampling generally reduces the ability of Tenax TA/Carbograph sorbent tubes to capture volatile compounds. However, since the impact was more pronounced for the less volatile compounds with higher breakthrough volumes, they speculated that impact is other than breakthrough. Ho et al. (2017) found significant impacts of RH on retention of $C_2$-$C_5$ aliphatic hydrocarbons in multisorbent (Tenax TA- Carbograph 1 TD- Carboxen 1003) tubes. The breakthrough volumes for $C_2$ aliphatic compounds were reduced by 13–22-fold under 90% RH and the main impact was expected to be through the less hydrophobic sorbent Carboxen 1003.  For >$C_5$ compounds no breakthrough was detected. In the studies of Maceira et al. (2017) humidity problems were demonstrated with carbon-based tubes, while Tenax-based tubes did not display any influence. Our tests with commercial tubes at 30 % humidity were done for the totally new tubes and this could also explain the higher retention. In the tests under 70 % humidity conducted after the tests of 30 % humidity, any difference between commercial and self-packed tubes was not that clear. However, tests with a ReGaS2 generator with commercial tubes at RH 30% conducted later show slightly higher breakthrough volumes, but not as high as for the first tests (Fig. 3), indicating some influence of RH. Increasing the temperature of the sorbent tube to 10 °C above the temperature of the air sample could reduce the interference of the humidity (Karbiwnyk et al. 2002).'

396:  … b) Recovery …..

**-corrected**

396:  What was used as reference (100%) for these tests?

**-100% was the expected amount in the NPL standard gas. To clarify this figure caption was changed to: 'Figure 4. a) Relative difference between the measured and expected amount in NPL gas standard with different online mode sampling times. b) Relative difference between the measured and expected amount of NPL gas standard flushed into different tubes (C=commercial, S=self-packed) or analyzed with online mode of the instrument (online). Error bars show the standard deviation between the three replicate samples. For both tests the instrument was calibrated with methanol solutions of BVOCs.'**

406-409:  This points towards difficulties and inconsistencies you had with the SQT analyses?  Please either detail and explain or maybe just drop SQT completely from the manuscript?

**-Actual analyses or calibrations were not a problem. The only problem was production of stable SQT containing air. Therefore, all other results and tests should be reliable also for them. Comment on methanol injection system on lines 406-409 was removed, since it is not clear that it**

**implies to the system producing terpene rich air and it is not the system we used for calibrating the instrument.**

408:  What do you actually really mean by 'online' sampling?  Sorry, that is still not really clear to me.

**-In online mode, sample is taken directly to the cold trap of thermal desorption unit without first concentrating it to the sorbent tube. We clarified this in the manuscript in section 2.2 in lines 153-154 (new version of the manuscript).**

464: Mention in figure caption that these are the results for the sodium thiosulfate ozone scrubber.

**-this was added**

473:  Pollmann

**-corrected**

499: This recommendation seems subjective. What really matters is the sampling volume through the filter and the particle loading.

**-This comment was removed from the manuscript.**

Figure 8:  These values are all positive.  Does that mean that all of the tested VOCs are artificially generated on the filters? Wouldn't one expect losses, i.e. negative numbers?

**-Most of these are still within uncertainties. The main observed differences come from the used stainless steel filter where the compounds from earlier measurements may have been released during the tests. For the other filters, only limonene shows high differences (around 5%) which may be due to its high reactivity and may be due to the sampling and analyses to a lesser extent.**

538: This really is an essential question that I would like to see addressed in much more depth:  How comparable are the liquid standard versus gas standard calibration results?

**-As discussed in section 3.3 and shown in figure 4a, deviation between liquid versus NPL gas standard calibration was less than 12 % for alpha-pinene, 3-carene and 1,8-cineol. However, compared to traceable certified NPL gas standard calibration, liquid calibration is prone to more errors and should be always compared to traceable gas standards if possible. Stable gas standards are available only for a few BVOCs and response of MS may vary between the BVOCs. Therefore, methanol standard was chosen for this study, where performance of the tubes were studied for many BVOCs. Lack of gas standards is also reason why comparison of liquid vs. gas standard was conducted only for a few compounds.**

**We tried to clarify this by modifying the sentences in section 3.6 on lines 570-574 (new version of the manuscript): 'Some differences between IMT and FMI were higher (23 % for α-pinene in Tenax TA) than the calculated uncertainty (7 % for both laboratories, Table 1, Supplement S2). A possible reason is the different calibration method and their associated uncertainties; FMI calibrated their TD-GC-MS with methanol solutions while IMT used a traceable NPL gas standard to calibrate their TD-GC-FID. Comparisons of liquid methanol standard and NPL gas standard in section 3.3 indicated that this can cause up to 12 % difference for α-pinene results.'**

Figure 9: Sampling volumes would be more meaningful than sampling times.

**-we corrected this to the figure**

Helmig, D. (1997), Ozone removal techniques in the sampling of atmospheric volatile organic trace gases, Atmospheric Environment, 31, 3635-3651.

**-the reference was added**

---

## Editor Decision (ED1)

Review of Hellén et al., Measurements of atmospheric C10-C15 biogenic volatile organic compounds (BVOC) with sorbent tubes', revised submission to AMT.

General:

The ms by Hellén et al. is focusing on the performance of adsorbent tubes and their analysis for the detection and quantification of C10-C15 BVOC. Adsorbent tubes have been the backbone of BVOC emission studies and even nowadays as 'online-techniques' (e.g. PTR-MS) got more and more available adsorbent tube sampling is important for compound specification (as e.g. PTR-MS can't distinguish MT species sharing the same mass), for use at sites with limited infrastructure or for process studies where the focus is not on longterm monitoring. Thus, detailed information on limitations and performance of these adsorbent tubes is urgently needed by the community especially for compounds which have not been in the focus before. In this sense this paper is an important contribution useful for a wide audience. The paper in general is well written with a clear structure and contains lots of useful information for the community that helps to improve sample setups and assess methodological challenges.

However, compared with the original submission, results and discussion of the breakthrough volumes has been improved, but I think this section will benefit from adding more information. If you aim to compare self-packed and commercially packed adsorbent tubes, a possible difference in breakthrough volumes could be caused by the mass of the sorbent in the tubes or its quality. Can you please add the mass of sorbent per tube to text and/or Table S1? Also, the breakthrough will depend (besides other) on the concentration of the gas sampled, and the authors have mentioned concentrations of the sample gas of 0.2 to 10 nmol/mol (L211). I suppose this range is caused by the different compounds, and the concentration used for each individual compound during the test have been constant? Then, comparing breakthrough volumes is rather misleading, as the amount of absorbed compound would depend both on volume and concentration of the sample air (amongst other). I would ask the authors to clarify this, e.g. by adding sample air concentration per compound to table 4, or adding the total absorbed mass per tube and compound.

Specific comments:

1) L13-17: I would recommend to change the order of sentences, e.g. have 'Even though online measurements…' first, then 'In this study…' to have a more logical order.
2) L16: '…online GC…'; shouldn't that be all online techniques here, e.g. even PTR-MS? The remaining part of the sentence is unclear to me. Please rephrase.
3) L19-22pp: I think it would be worth stating in the abstract that no multibed configurations have been tested here.
4) L31: quantify here, how good was the agreement between the two labs?
5) L52: one '(' too much.
6) Ch 2.4, breakthrough tests: tell how many tubes were used for these experiments.
7) L319: insert 'were' between filters' and 'inserted'

8) Tab3: 'new filter that has been used…' is a bit awkward I think, what about 'aged or used filter' instead. Tell how long it had been used before and for which kind of measurements (low/medium/high concentrations).
9) L355: '…TA tubes both stored at…'
10) L358: significant at which level?
11) L395: 'In earlier studies the impact…'
12) L401: what is a x-fold reduction? Reduced to y %?
13) Fig2: Any possibility to give the numbers of samples here? At least in the caption (N=x for …)
14) L402: 'For >C5 components …was detected'. By Ho et al.? Unclear what you are referring to here.
15) L426: 'In tubes on Tenax TA…', connect with next sentence!

---

## Author Response (AR2)

Dear editor,

We have carefully considered all the comments by the reviewer and improved the manuscript based on them as explained in the following sections.

Yours sincerely,

Heidi Hellén

**Answers to the reviewer comments:**

The manuscript "Measurements of atmospheric C10-C15 biogenic volatile organic compounds (BVOCs) with sorbent tubes" by Hellen et al., investigates different materials, sampling efficiency and set-up used for measuring terpenoid compounds using sorbent tubes. I find the manuscript important for the scientific community, specifically for the many groups conducting these measurements and for those willing to start them. The manuscript presents several novel aspects, the methods used are robust and presented in an exhaustive way. I highly recommend the manuscript for publication, and only suggest the authors to provide a few more details that could help other researchers to follow the reported procedures:

L. 118. A brief description of the self-packing procedure is presented at the end of the section. Can the authors provide more details about the procedure, amount of the material used, cleaning processes, conditioning times after the tubes have been assembled? This would be a valuable information to add in the SI since many groups are packing the tubes by themselves but procedures may differ between groups. Are there standard methods reported in literature that could be mentioned here?

*-The tubes were packed using a gauze loading rig (PerkinElmer) and following the instructions by Perkin Elmer (Technical note: Packing Thermal Desorption Sample Tubes, 2007). First a stainless-steel net was inserted into the grooved end of a tube. The tube was filled with sorbent leaving ~1.5 cm free from the other end of the tube. Using the gauze loading rig a stainless steel net was inserted also into this end of the tube and after that a spring was adjusted into the tube to keep the net and sorbents fixed. The tubes were cleaned following the instructions provided by sorbent producers. Cleaned tubes were analysed before the use to verify that they were free of contamination. We added a reference to the packing instructions into the manuscript (L- 119-120).*

L. 147. Did the authors try different desorbing flows and times that could be mentioned here?

*- in this study we did not test different desorption flows since used desorption method was found to be efficient*

L. 148. For how long was the sample focused into the cold trap?

*-we added to the manuscript that tubes were desorbed for 5 minutes (L. 148).*

L. 154. Was there a particular set-up used for liquid injections and which repeatability between different injections of the different compounds was achieved throughout a calibration?

*-there was no particular set-up. Just a self-constructed system with stainless steel tubing and a T-connector where solution was injected through a septum into the sorbent tube. Repeatability is included into the uncertainty of the method, and is shown for the alpha-pinene, beta-pinene and*

*toluene in supplement S2. For Tenax TA tubes repeatability was 2-4 % being highest for terpinolene. We added a comment on this into the manuscript (L. 159).*

L. 165. Is there any explanation for the behavior of toluene and was this observed also for similar compounds (e.g. benzene and benzaldehyde)?

*-yes, there is always some background for benzene, toluene and benzaldehyde in Tenax TA tubes. A possible source could be the breakdown of sorbents.*

L. 274. How long was the aging time of the scrubbers?

*-aging depends on the scrubber, ozone levels and used flows. In this study stainless steel and $Na_2S_2O_3$ scrubbers were aged for 10 days at an ozone level of ~44 nmol/mol in the flows of 0.1 L $min^{-1}$ and 0.07 Lmin-1, respectively. $MnO_2$ scrubbers were aged in a flow of humidified air (1 L/min) enriched with 120 nmol/mol of $O_3$ for 216 h.  KI/Cu scrubbers were installed at a French EMEP rural site "Peyrusse Vieille", for the measurement of oxy-VOCs with DNPH cartridges: 2 measurements/week at a flow rate of 1L/min for 4 hours (from 12h-16h UTC) for one month and a half. The ozone concentrations ranged from 40 to 60 nmol/mol. Ozone removal capacities of the scrubbers have been studied earlier (e.g. Hellén et al. 2012) and should be tested for specific experiments. We added a description on aging processes into the manuscript Table 2.*

L. 517. Did the author investigate any O3 concentration dependency?

*-not in this study*

Figure 8. How much aerosol mass was loaded on the filter? And how aged was the F used?
*We didn't measure the aerosol's mass loaded. The selected filter was used during a one-month field campaign.*
SI. S2. Did the authors assume that the chromatograms integration was very good or was this "good" assessed somehow?

*-It was not specifically assessed since separation of the peaks was so clear.*

---

## Author Response (AR3)

Dear editor,

We have carefully considered the comments by the additional reviewer and improved the manuscript based on them as explained in the following sections.

Yours sincerely,

Heidi Hellén

**Answers to the reviewer comments:**

General:

The ms by Hellén et al. is focusing on the performance of adsorbent tubes and their analysis for the detection and quantification of C10-C15 BVOC. Adsorbent tubes have been the backbone of BVOC emission studies and even nowadays as 'online-techniques' (e.g. PTR-MS) got more and more available adsorbent tube sampling is important for compound specification (as e.g. PTR-MS can't distinguish MT species sharing the same mass), for use at sites with limited infrastructure or for process studies where the focus is not on longterm monitoring. Thus, detailed information on limitations and performance of these adsorbent tubes is urgently needed by the community especially for compounds which have not been in the focus before. In this sense this paper is an important contribution useful for a wide audience. The paper in general is well written with a clear structure and contains lots of useful information for the community that helps to improve sample setups and assess methodological challenges.

However, compared with the original submission, results and discussion of the breakthrough volumes has been improved, but I think this section will benefit from adding more information. If you aim to compare self-packed and commercially packed adsorbent tubes, a possible difference in breakthrough volumes could be caused by the mass of the sorbent in the tubes or its quality. Can you please add the mass of sorbent per tube to text and/or Table S1? Also, the breakthrough will depend (besides other) on the concentration of the gas sampled, and the authors have mentioned concentrations of the sample gas of 0.2 to 10 nmol/mol (L211). I suppose this range is caused by the different compounds, and the concentration used for each individual compound during the test have been constant? Then, comparing breakthrough volumes is rather misleading, as the amount of absorbed compound would depend both on volume and concentration of the sample air (amongst other). I would ask the authors to clarify this, e.g. by adding sample air concentration per compound to table 4, or adding the total absorbed mass per tube and compound.

-We do not have the exact masses of the sorbents in the self-packed tubes. In the commercial tubes amount is usually ~200 mg/tube and for self-packed tubes similar length of the sorbent bed was used and therefore amounts are expected to be similar.

-The used amount fractions in the sampling air with methanol solution tests were very low (0.1-.0.9 nmol/mol). In the tests with ReGaS2 permeation system higher amount fractions (~10 nmol/mol) were used. This is now clarified in the manuscript on lines 203-207. Constant concentrations of the individual compounds were used during each test, but there were some variations between the

tests. Amount fractions used in each test were added to the table 4. We assumed that for the low amount fractions, breakthrough is not strongly dependent on the amount fraction, but more on the flushed volume of air. However, we added a comment to the manuscript that also, a bit lower amount fractions used for the commercial tubes in the tests with methanol solutions may have had an impact (l. 392-393).

Specific comments:

1) L13-17: I would recommend to change the order of sentences, e.g. have 'Even though online measurements…' first, then 'In this study…' to have a more logical order.

-we did this (L. 13-17)

2) L16: '…online GC…'; shouldn't that be all online techniques here, e.g. even PTR-MS? The remaining part of the sentence is unclear to me. Please rephrase.

-this is true. We removed the word 'GC'. (L. 14). We rephrased the sentence into 'Even though online measurements of BVOCs are becoming more common, the use of sorbent tubes is expected to continue because they offer greater spatial coverage compared to online measurements, and no infrastructure (e.g., electricity, housing/shelter with stable temperature and humidity, sampling lines) is required for sampling.'

3) L19-22pp: I think it would be worth stating in the abstract that no multibed configurations have been tested here.

-we added to the line 22 'No multibed configurations were tested.'

4) L31: quantify here, how good was the agreement between the two labs?

-since there were variable results for different tests and compounds, which would need explanations on the differences, we are not able to shortly quantify it here.

5) L52: one '(' too much.

-corrected (L. 52)

6) Ch 2.4, breakthrough tests: tell how many tubes were used for these experiments.

-we did not list the number of different tubes used. We used randomly the tubes, which were used also in the other tests and are listed in the Table S1.

7) L319: insert 'were' between filters' and 'inserted'

-done (L. 301)

8) Tab3: 'new filter that has been used…' is a bit awkward I think, what about 'aged or used filter' instead. Tell how long it had been used before and for which kind of measurements (low/medium/high concentrations).

We changed the text in removing "new" (Table 3). We didn't measure the aerosol's mass loaded. The selected filter was used during a one-month field campaign (medium to high concentrations).

9) L355: '…TA tubes both stored at…'

-corrected (L. 357)

10) L358: significant at which level?

-we do not have statistical data on the significance. We changed the wording to 'clear' (L 340).

11) L395: 'In earlier studies the impact…'

-corrected (L. 377)

12) L401: what is a x-fold reduction? Reduced to y %?

-Corrected to 'reduced by 92 - 95%' (L. 383)

13) Fig2: Any possibility to give the numbers of samples here? At least in the caption (N=x for …)

-We added into figure caption '(N = 4, 5, 3 and 5 for Tenax TA stainless steel commercial, Tenax TA stainless steel self-packed, Tenax TA SilcoNert 1000 self-packed and Carbopack B commercial, respectively)'

14) L402: 'For >C5 components …was detected'. By Ho et al.? Unclear what you are referring to here.

-we added a reference to Ho et al. (L 384)

15) L426: 'In tubes on Tenax TA…', connect with next sentence

-corrected  (L410-411)